# Secondary-structure switch regulates the substrate binding of a YopJ family acetyltransferase

Yao Xia[1,2,8], Rongfeng Zou[3,8], Maxime Escouboué[4,8], Liang Zhong[1,2,8], Chengjun Zhu[1,2], Cécile Pouzet[5], Xueqiang Wu[6], Yongjin Wang[1,2], Guohua Lv[7], Haibo Zhou [6], Pinghua Sun [1,2✉], Ke Ding [1,2✉], Laurent Deslandes [4✉], Shuguang Yuan[3✉] & Zhi-Min Zhang [1,2✉]

The Yersinia outer protein J (YopJ) family effectors are widely deployed through the type III secretion system by both plant and animal pathogens. As non-canonical acetyltransferases, the enzymatic activities of YopJ family effectors are allosterically activated by the eukaryote-specific ligand inositol hexaphosphate (InsP6). However, the underpinning molecular mechanism remains undefined. Here we present the crystal structure of apo-PopP2, a YopJ family member secreted by the plant pathogen *Ralstonia solanacearum*. Structural comparison of apo-PopP2 with the InsP6-bound PopP2 reveals a substantial conformational readjustment centered in the substrate-binding site. Combining biochemical and computational analyses, we further identify a mechanism by which the association of InsP6 with PopP2 induces an α-helix-to-β-strand transition in the catalytic core, resulting in stabilization of the substrate recognition helix in the target protein binding site. Together, our study uncovers the molecular basis governing InsP6-mediated allosteric regulation of YopJ family acetyltransferases and further expands the paradigm of fold-switching proteins.

[1] International Cooperative Laboratory of Traditional Chinese Medicine Modernization and Innovative Drug Development of Chinese Ministry of Education (MOE), College of Pharmacy, Jinan University, 510632 Guangzhou, China. [2] Guangdong Province Key Laboratory of Pharmacodynamic Constituents of TCM and New Drugs Research, College of Pharmacy, Jinan University, 510632 Guangzhou, China. [3] Shenzhen Institutes of Advanced Technology, Chinese Academy of Sciences, 518005 Shenzhen, China. [4] Laboratoire des Interactions Plantes-Microbes-Environnement (LIPME), INRAE, CNRS, Université de Toulouse, 31326 Castanet-Tolosan, France. [5] FRAIB-TRI Imaging Platform Facilities, FR AIB, Université de Toulouse, CNRS, 31320 Castanet-Tolosan, France. [6] Institute for Pharmaceutical Analysis, College of Pharmacy, Jinan University, 510632 Guangzhou, China. [7] Division of Histology & Embryology, Medical College, Jinan University, 510632 Guangzhou, China. [8]These authors contributed equally: Yao Xia, Rongfeng Zou, Maxime Escouboué, Liang Zhong. ✉email: pinghuasunny@163.com; dingke@jnu.edu.cn; laurent.deslandes@inrae.fr; shuguang.yuan@siat.ac.cn; 13632107756@163.com

To establish successful infection, most Gram-negative bacteria depend on a specialized macromolecular syringe, the type III secretion system (T3SS), to transport effector proteins into host cells[1]. These type III–secreted effectors (T3SEs) modulate a variety of host cell signaling pathways, especially those involved in immune response, thereby facilitating the establishment of an environmental niche for the pathogens in which to thrive[2]. Owing to the rapid co-evolutionary arms race between pathogens and their hosts, T3SE repertoires are highly variable in different pathogen species. One notable exception is the Yersinia outer protein J (YopJ) effector family, whose members are uniquely produced by a wide variety of bacterial pathogens, including animal pathogens *Yersinia* spp., *Salmonella enterica*, *Vibrio parahaemolyticus*, and *Aeromonas salmonicida*, and the plant pathogens *Pseudomonas syringae*, *Xanthomonas campestris*, *Ralstonia solanacearum*, *Erwinia amylovora*, and *Acidovorax citrulli*[3]. This conservation suggests a critical virulence role of the YopJ family effectors during host colonization and dissemination.

All the YopJ family effectors contain a catalytic core, which is structurally similar to that of the CE clan of cysteine proteases, with a conserved histidine–aspartate/glutamate–cysteine catalytic triad[4]. However, mounting evidence has demonstrated that YopJ family effectors possess acetyltransferase activity. For instance, YopJ family effectors produced by mammalian pathogens, including YopJ from *Yersinia* spp. and AvrA from *S. enterica*, target the mitogen-activated protein kinase and/or nuclear factor-κB signaling pathways by acetylating specific serine and threonine residues in the activation loop of the targeted kinases. This acetylation blocks the phosphorylation and activation of these kinases, leading to a suppressed inflammatory response and cell death of immune-related cells[5–10]. Interestingly, the YopJ effectors share no sequence similarity with any other representative acetyltransferases such as the well-studied N-terminal acetyltransferases[11] and histone acetyltransferases[12], suggesting a distinctive enzymatic mechanism that might be adopted by the YopJ family of acetyltransferases[3]. Furthermore, the target proteins of YopJ acetyltransferases produced by plant pathogens are highly diverse. Among them, PopP2 produced by the root-infecting bacterium *R. solanacearum* acetylates a conserved lysine residue in WRKY transcription factors to inhibit the expression of defense-related genes[13,14]. In Arabidopsis, PopP2 is recognized by a pair of plant immune receptors, RPS4 (Resistance to Pseudomonas syringae4) and RRS1-R (Resistance to Ralstonia solanacearum1). RRS1-R contains at its carboxyl terminus a conserved WRKY DNA-binding domain that acts as a decoy to detect PopP2 interference with defensive WRKY transcription factors[13,14]. Acetylation of RRS1-R WRKY domain by PopP2 inhibits its DNA-binding activity and triggers activation of the RPS4/RRS1-R pair, which results in activation of plant immunity, whereas HopZ1a secreted by *P. syringae* interacts with a number of target proteins, including tubulin[15], Jasmonate ZIM domain (JAZ) proteins[16] and hydroxyisoflavanone dehydratase (GmHID1)[17].

Another striking difference between the YopJ family members and other acetyltransferases is that the activation of YopJ family effectors relies on the host cofactor inositol hexaphosphate (InsP6)[18], which is abundant in most eukaryotic cells but absent in bacteria. Our recent studies identified a previously undescribed regulatory domain in the structures of PopP2[19] and HopZ1a[20]. Both the binding pockets of InsP6 and the acetyl group donor acetyl-coenzyme A (AcCoA) are located on the interface between the regulatory domain and the catalytic core. Biochemical and computational studies support that InsP6 binding to the YopJ family effectors stabilizes the conformation of the regulatory domain, thus forming the AcCoA-binding pocket near the catalytic center. Further study on PopP2 revealed that the PopP2–WRKY association is also InsP6 dependent[19]. However, we did not observe extensive interactions between the regulatory domain and WRKY, indicative of an unknown mechanism underlying the InsP6-regulated substrate binding.

Here we present the crystal structure of PopP2 in apo-state, which reveals a large structural change from our previously reported PopP2-InsP6 complexes. We further demonstrate that the interaction of InsP6 with PopP2 triggers an intricate cascade of conformational changes. Importantly, InsP6 regulates substrate binding of PopP2 by inducing a helix-to-strand fold switching in the catalytic core, thereby allosterically stabilizing the substrate-interacting α-helix. Our study provides critical mechanistic insights into the allosteric regulation of YopJ family acetyltransferases.

## Results

**The crystal structure of PopP2 in apo-state.** To understand how InsP6-binding induces the structural change of YopJ effectors, we crystallized the entire acetyltransferase domain (residues 149–488) of PopP2 in the absence of InsP6. The crystal structure was solved to a resolution of 2.3 Å via molecular replacement using the catalytic core of the PopP2-InsP6 complex structure[19] as a search model (Supplementary Table 1). The apo-state PopP2 structural model consists of a five-stranded β-sheet sandwiched by αD, αE, and αF from one side and αC from the other side and a helix-bundle formed by the αB and αA from the very N-terminus and αG and αH from the very C-terminus (Fig. 1). In our previous studies, αA, αG, and αH were considered as part of the regulatory domain. In light of the fact that this helix-bundle exists in both apo and complex structures and interacts tightly with the central β-sheet to form a rigid body, we redefined them as components of the catalytic core. The catalytic triad, including His260, Asp279, and Cys321 residues, are well preserved in the structure (Fig. 1b). On the other hand, the entire regulatory domain (residues 377–447) was unable to build due to its missing electron density.

**Structural comparison between apo- and InsP6-bound PopP2 structures.** Structural comparison of the apo-PopP2 structure with that of the InsP6-bound PopP2 reveals that the catalytic core is well conserved, giving a root mean square deviation (RMSD) of 0.5 Å over the Cα atom of 248 aligned residues (Fig. 2a). Nevertheless, we observed three obvious structural differences, besides the flexible regulatory domain. The most striking difference lies in a segment ahead of the N-terminus of the regulatory domain (residues 351–375), which is able to adopt two entirely different structures with distinct sets of packing interactions (Fig. 2b). In the apo conformation, this segment exists as a long α-helix (αF) that is packed against αE and β5 mainly through hydrogen bonds and hydrophobic interactions; by contrast, in the PopP2-InsP6 complex structure, the same set of residues are folded into two antiparallel β-strands (β6' and β7') linked by a long loop ($L_{\beta6'\beta7'}$). Notably, β6' and β7' in the complex structure further join the β-strands of the catalytic core to form a seven-stranded β-sheet (Fig. 1c). This observation suggests that this segment may undergo a local reshuffling of secondary structure induced by InsP6 binding. Indeed, it is apparent that αF in apo-PopP2 would sterically clash with the first helix of the regulatory domain, which harbors R380 and K383 residues that directly interact with the phosphate groups of InsP6 (Supplementary Fig. 1). Therefore, to accommodate InsP6 binding, the segment L371-R380 needs to push αF away, which may lead to the fold switching of αF.

Another prominent structural change occurs in a fragment spanning residues 287–302 (Fig. 2c). In the apo structure, only a short loop ($L_{\beta4\beta5}$) containing residues 296–302 was built with poor electron density. This loop appears loosely constrained, with

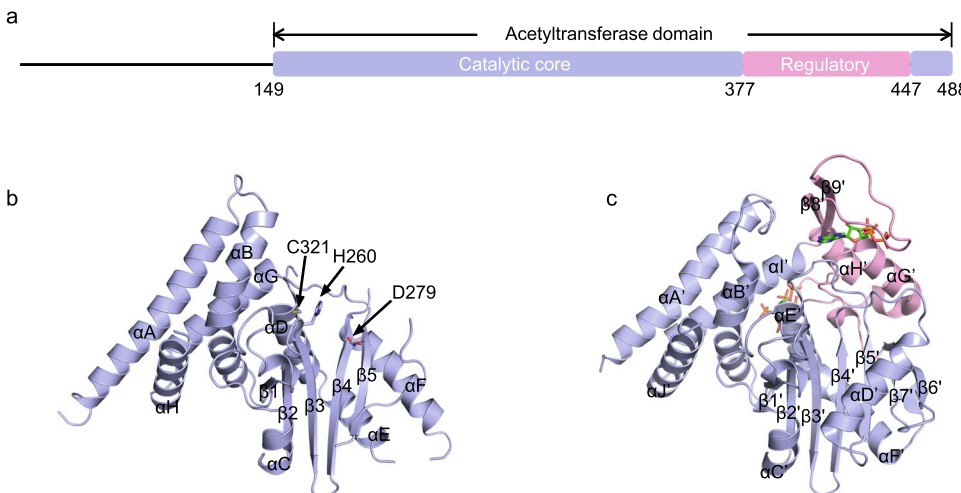

**Fig. 1 Structures of PopP2 in apo and InsP6-bound state. a** Schematic illustration of PopP2 protein, with the regulatory domain in pink and the catalytic core in blue. Similar colors are used in other panels unless otherwise indicated. **b** Crystal structure of PopP2 in apo state. The α-helices and β-strands are counted in alphabetic and numeric orders, respectively. The catalytic triad (H260/D279/C321) is shown as a stick representation. The regulatory domain (residues 377–447) is absent. **c** Crystal structure of PopP2 in complex with InsP6 and CoA (PDB code: 5W3Y). InsP6 and CoA are shown in stick presentation.

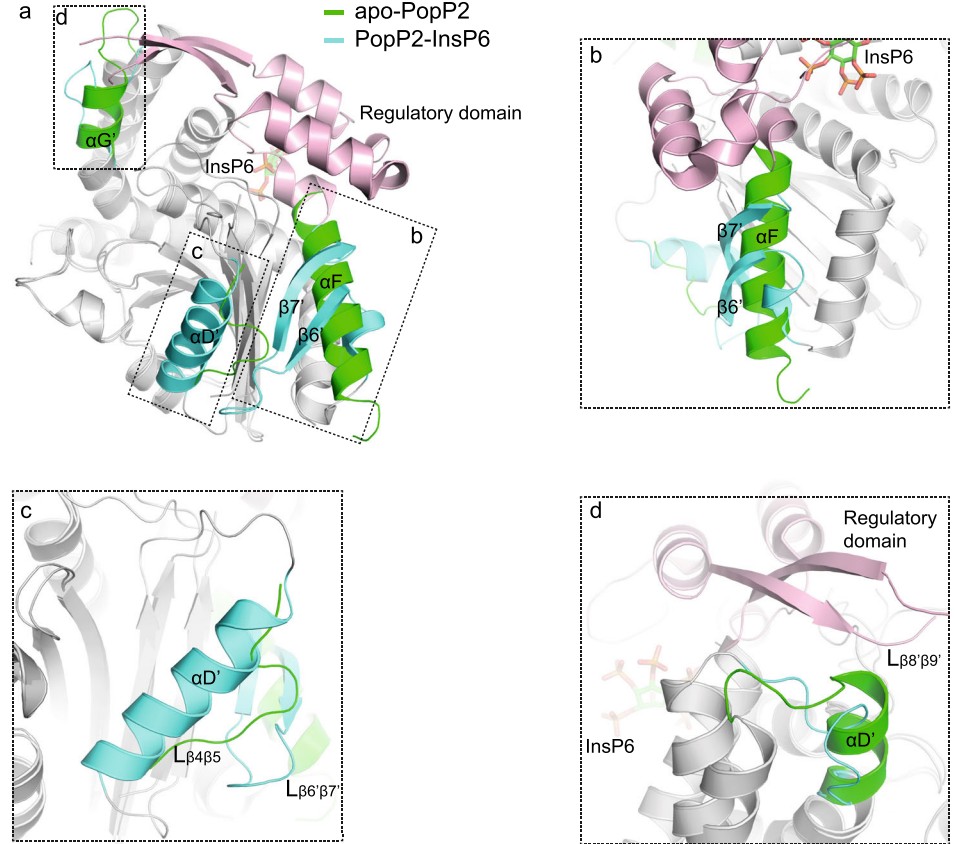

**Fig. 2 Structural comparison of the apo- and InsP6-bound PopP2. a** Overlap of the structures of the apo- and InsP6-bound PopP2. The well-overlapped catalytic core regions of both apo- and InsP6-bound PopP2 are colored in gray. The regulatory domain is colored in pink. Structures with obvious differences between the apo- and InsP6-bound PopP2 structures are colored in green for the apo-PopP2 and in cyan for the InsP6-bound PopP2, respectively. The major differences are highlighted in expanded view in **b**–**d**.

lack of a significant interaction either within the loop or with the β-sheet in the catalytic core. In the presence of InsP6, this fragment folds into an α-helix (αD'). Importantly, extensive contacts were observed between αD' and $L_{\beta6'\beta7'}$ in the fold-

switching region, both of which are directly involved in the interaction with the WRKY DNA-binding domain of the Arabidopsis RRS1-R immune receptor, hereafter designated as RRS1$_{WRKY}$[19]. RRS1$_{WRKY}$ domain was previously shown to act as

an effector target "decoy" whose acetylation by PopP2 triggers activation of the RPS4/RRS1-R-dependent immunity[13,14]. The last obvious conformational change arises from αG in the helix-bundle. αG exists in both the apo- and InsP6-bound structures, although it appears shorter in the InsP6-bound structure. The C-terminus of αG is too close to the $L_{\beta8'\beta9'}$ of the regulatory domain and RRS1$_{WRKY}$ in the overlapped structures, suggesting that the stabilization of the regulatory domain by InsP6 leads to disruption of the C-terminal helical structure of αG to save room for substrate binding (Fig. 2d and Supplementary Fig. 2). Together, the structural comparison reveals three different types of structural transformations triggered by a single binding event: fold switching, disorder to order, and order to disorder. And these transformations occur on PopP2 globally, both in the catalytic core and the regulatory domain.

**InsP6 triggers fold switching of PopP2.** To test whether the fold-switching motif (residues 351–375) exhibits different folds in the crystal and in solution, we collected Raman spectra of PopP2 in the absence or presence of InsP6, using surface-enhanced Raman scattering technique (Fig. 3a). Peaks observed in the experiment are the reflection of those secondary structures close to the gold nanoparticles (Au@Ag NPs) that bind with the protein. The result showed that apo-PopP2 has an obvious peak at 1657 cm$^{-1}$, which attributes to the C=O stretching vibration in the amide I band of α-helix[21]. However, this peak disappeared after incubation with InsP6, indicative of reshuffling of some α-helical structures induced by InsP6. We find several amide groups (Asn 296, Asn298, Asn348, and Gln360) and one thiol group (Cys307) near or on αF that are exposed to the solvent and maybe helpful to bind the nanoparticles. It is highly possible that the peak at 1657 cm$^{-1}$ is contributed by the αF–Au@AgNP interactions.

To provide mechanistic insight into the InsP6-mediated fold-switching process of PopP2, we performed metadynamics simulation, which is widely used for sampling biologically rare events[22]. We introduced path collective variables (path-CVs) to sample how the fold-switching motif is transited from an α-helix into a β-hairpin (Fig. 3b and Supplementary Movie 1). As shown in Fig. 3b, two minima were identified, corresponding to the apo- and InsP6-bound structures, respectively. There is ~3 kcal mol$^{-1}$ difference in the free energies between these two minima, with the apo structure being the global minimum, suggesting that the fold-switching motif assumes a helix-dominant conformation in the absence of InsP6, which is in good agreement with what is observed in the crystal structure. During the transition process, the fold-switching region turns into a disordered loop. Therefore, the helix in the fold-switching region needs to unfold upon InsP6 binding and then refold into antiparallel β-strands.

**InsP6 regulates substrate binding through the fold-switching motif.** Since the InsP6 binding is necessary for substrate binding of PopP2 and the fold-switching motif bridges the InsP6 binding pocket and the substrate recognition helix αD', we hypothesized that InsP6 might regulate substrate binding through inducing secondary structure shuffling to stabilize the substrate recognition helix. To test this possibility, we first performed $3 \times 500$ ns all-atom molecular dynamics (MD) simulations for apo- and InsP6-bound PopP2, respectively. The principle component analysis on the MD trajectories indicated that apo-PopP2 contains a much more flexible RRS1$_{WRKY}$-binding region than that of the InsP6-bound PopP2 (Supplementary Fig. 3a–d and Supplementary Movies 2 and 3), with an average RMSD value of 5.8 Å for apo-PopP2 and 4.0 Å for InsP6-bound PopP2 during the final 100 ns MD simulations, respectively. Due to the fold switch induced by InsP6, a large loop ($L_{\beta4\beta5}$ in apo-PopP2) interacts with the fold-switching motif and refolds into αD', leading to a relatively smaller and deeper pocket, which may facilitate stronger substrate binding (Supplementary Fig. 3e, f). Molecular mechanics/generalized Born surface area (MM/GBSA) method was used to calculate binding energies of RRS1$_{WRKY}$ to the InsP6-bound PopP2 and the apo-PopP2. Consistently, it demonstrated that RRS1$_{WRKY}$ has a much more favorable binding energy in the InsP6-bound PopP2 over that of apo-PopP2: $-9.1 \pm 4.3$ vs $-3.8 \pm 3.8$ kcal mol$^{-1}$, respectively (Supplementary Table 2).

We further performed bio-layer interferometry (BLI) assays and acetylation assays to investigate how residues in the InsP6-binding pocket (R380 and K383) and fold-switching motif affect the interaction between PopP2 and RRS1$_{WRKY}$. In the InsP6-binding pocket, we selected two residues Arg380 and Lys383, both located on a short helix immediately following the fold-switching motif, and mutated them to Alanine (R380A and K383A mutants, respectively). To disrupt signal transition between the InsP6-binding pocket and substrate recognition helix, we generated two PopP2 mutants in the fold-switching motif, L369P/V370P and L371P/D372P, each containing on β7' (αF in the apo structure) a pair of point mutations that replaced linear side chains with Proline (Supplementary Fig. 4). Given that PopP2 was reported to have autoacetylation activity when overexpressed in *Escherichia coli*[23], we first investigated the acetylation level of purified PopP2 proteins (Supplementary Fig. 5). While wild-type (WT) and R380A mutant showed strong acetylated Lys (AcK) signal by immunoblot, no signal could be detected with the other mutants, indicating that they were all impaired in their autoacetylation activity. We then used RRS1$_{WRKY}$ as a PopP2 substrate to carry out acetylation assays in *E. coli*. (Fig. 4a). Consistently, both WT PopP2 and R380A mutant acetylated RRS1$_{WRKY}$ robustly (Fig. 4a), thus disrupting the association between RRS1$_{WRKY}$ and DNA (Fig. 4b). We did

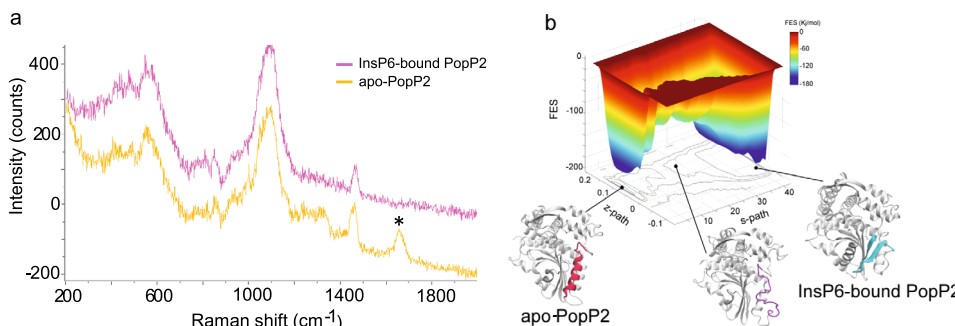

**Fig. 3 InsP6 induces fold switch of PopP2.** **a** Raman spectra of PopP2 in the absence (yellow) or presence (pink) of InsP6. **b** Metadynamics simulation result of the fold-switching process of PopP2. Free energy profile acquired from metadynamics simulation is shown on the top, "s-path" means progress along the path, "z-path" means deviation along the path.

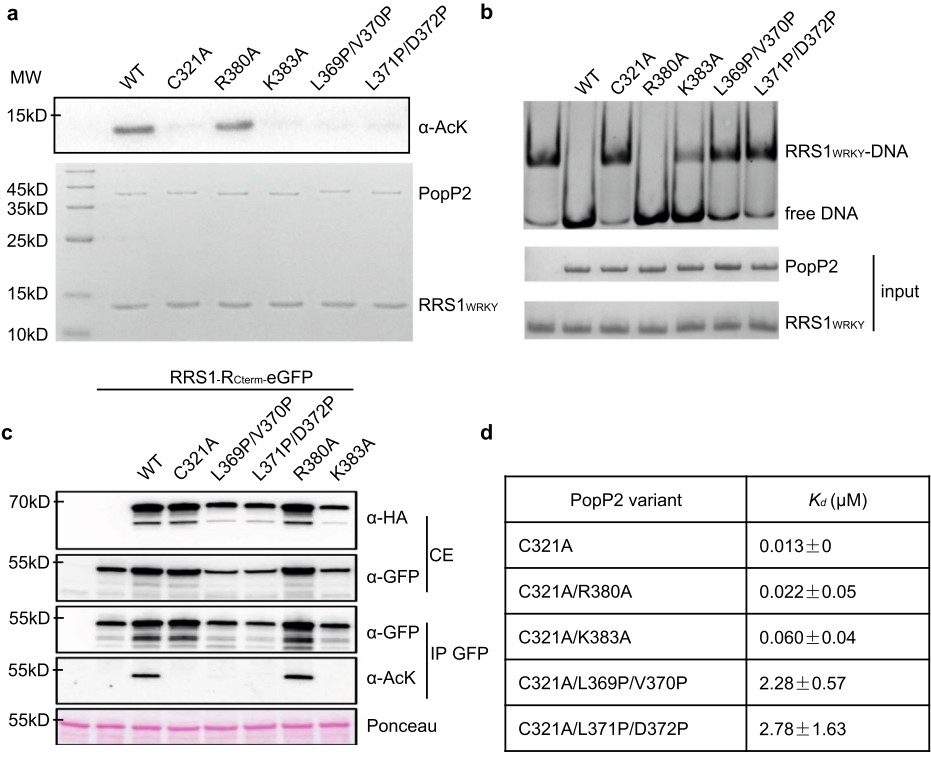

**Fig. 4 InsP6 regulates RRS1$_{WRKY}$ binding via the fold switch of PopP2. a** In vitro acetylation assay of PopP2 variants in the presence of InsP6 and AcCoA, with RRS1$_{WRKY}$ as substrate. This experiment was performed twice with similar results. The protein amount is indicated by Coomassie blue staining (bottom). **b** Electrophoretic mobility shift assay of PopP2 variants in the presence of InsP6, AcCoA, and W-box DNA, with RRS1$_{WRKY}$ as substrate. Only unacetylated RRS1$_{WRKY}$ proteins form complex with W-box DNA. The protein–DNA complexes and free DNA were detected by ethidium bromide (top), and the protein amount is indicated by Coomassie blue staining (bottom). **c** Immuno-detection of acetylated RRS1-R C-terminal portion in planta in the presence of WT PopP2 and PopP2-R380A mutant. Transient expression of 3HA-tagged PopP2 variants and eGFP-tagged RRS1-R$_{Cterm}$ were performed in *N. benthamiana* leaves, with samples harvested at 48 hpi. Detection of HA- and eGFP-tagged proteins was conducted using anti-HA and anti-GFP antibodies, respectively. Ponceau S staining of total proteins indicates equal loading of the samples. This experiment was conducted three times with similar results. **d** Binding analysis results of WT and PopP2 variants. The $K_d$ represents the mean value of two independent experiments.

not observe apparent acetylation of RRS1$_{WRKY}$ by K383A, but it led to modest reduction to the DNA binding of RRS1$_{WRKY}$, suggesting that this mutant may still have weak acetyltransferase activity. The L369P/V370P and L371P/D372P mutations completely abolished acetyltransferase activity (Fig. 4a, b). Consistent with our in vitro acetylation data, similar results were observed in *Nicotiana benthamiana* leaves upon transient co-expression of 3HA-tagged PopP2 variants with the 189-amino acid C-terminal portion of RRS1-R containing the WRKY domain (position 1190–1378, hereinafter called RRS1R$_{C-term}$) fused to the enhanced green fluorescent protein (RRS1R$_{C-term}$-eGFP) (Fig. 4c).

BLI analysis performed on recombinant proteins revealed that, of the two mutants in InsP6-binding pocket, only K383A reduced the binding affinity of RRS1$_{WRKY}$ to PopP2 by about five times (Fig. 4d and Supplementary Fig. 6). PopP2 has previously been shown to autoacetylate on K383[23], thus disrupting the interaction between InsP6 and PopP2. Therefore, to circumvent this problem in BLI assays, the R380A, K383A, and the two di-mutations were introduced in the sequence of catalytically inactive PopP2-C321A mutant. Interestingly, both double mutations significantly decreased the binding affinity of PopP2 to RRS1$_{WRKY}$ in the presence of InsP6 (Fig. 4d and Supplementary Fig. 6). Together, these data support the idea that the fold switching induced by InsP6 binding is important for PopP2 substrate recognition.

**PopP2 substrate targeting in planta is compromised by mutations in the fold-switching motif.** To further confirm the

importance of PopP2 fold-switching motif for substrate targeting in plant cells, we performed a FRET-FLIM (Förster resonance energy transfer by fluorescence lifetime imaging) assay that was previously used successfully for detection of interactions between PopP2 and RRS1-R in the nucleus[23]. Here the different PopP2 variants were C-terminally fused to the cyan fluorescent protein (CFP) to serve as FRET donor and transiently expressed in *N. benthamiana* either alone or with RRS1-R$_{Cterm}$ fused to the Venus variant of the yellow fluorescent protein (YFPv; serving as a FRET acceptor). The average CFP lifetime in nuclei expressing PopP2-CFP alone was 2.8377 ± 0.01290 ns (mean ± SEM). A significant reduction of the average CFP lifetime to 2.5356 ± 0.03059 ns (FRET efficiency of 10.65%, $p$ value = 3.42217E−15) was measured in the nuclei co-expressing the PopP2-CFP and RRS1-R$_{Cterm}$-YFPv fusion proteins (Table 1 and Supplementary Fig. 7), demonstrating that PopP2 interacts with its substrate in vivo. PopP2-R380A was also found to interact with RRS1-R$_{Cterm}$, but to a lesser extent than WT PopP2, as evidenced by the reduction of the average CFP lifetime in the nuclei co-expressing R380A-CFP and RRS1-R$_{Cterm}$-YFPv, compared with the nuclei expressing R380A-CFP alone (Table 1 and Supplementary Fig. 7). By contrast, for the other PopP2 mutants (L369P/V370P, L371P/D372P, and K383A), no interaction could be detected with RRS1-R$_{Cterm}$, consistent with the data shown in Figs. 3 and 4. Unexpectedly, an increase in their CFP lifetime was monitored in the presence of RRS1-R$_{Cterm}$-YFPv, probably as a consequence of a change in the nuclear environment of PopP2 triggered by RRS1-R$_{Cterm}$ (Table 1 and Supplementary Figs. 7 and 8).

**Table 1 FRET-FLIM measurements showing that PopP2 L369P/V370P, L371P/D372P, and K383A are affected in their ability to physically interact in planta with the C-terminal portion of RRS1-R.**

| Donor | Acceptor | T (ns)[a] | $\Delta t$ (ns)[b] | Sem[c] | N[d] | E (%)[e] | p value[f] |
|---|---|---|---|---|---|---|---|
| PopP2-CFP | — | 2.8377 | 0.302 | 0.01290 | 60 | | |
| PopP2-CFP | RRS1-R$_{Cterm}$-YFPv | 2.5356 | | 0.03059 | 61 | 10.65 | 3.42E−15 |
| L369P-V370P-CFP | — | 2.5750 | −0.185 | 0.02366 | 40 | | |
| L369P-V370P-CFP | RRS1-R$_{Cterm}$-YFPv | 2.7600 | | 0.02483 | 40 | −7.18 | 7.23E−07 |
| L371P-D372P-CFP | — | 2.5353 | −0.062 | 0.02911 | 59 | | |
| L371P-D372P-CFP | RRS1-R$_{Cterm}$-YFPv | 2.5968 | | 0.03179 | 60 | −2.43 | 0.15615099 |
| R380A-CFP | — | 2.7649 | 0.098 | 0.02189 | 39 | | |
| R380A-CFP | RRS1-R$_{Cterm}$-YFPv | 2.6672 | | 0.02571 | 39 | 3.53 | 0.004988117 |
| K383A-CFP | — | 2.5619 | −0.241 | 0.04220 | 40 | | |
| K383A-CFP | RRS1-R$_{Cterm}$-YFPv | 2.8028 | | 0.02391 | 40 | −9.40 | 3.94E−06 |

[a]Mean lifetime, $T$, in nanoseconds (ns). For each nucleus, average fluorescence decay profiles were plotted and fitted with exponential function using a non-linear square estimation procedure and the mean lifetime was calculated according to $T = \Sigma\alpha_i t_i^2 / \Sigma\alpha_i t_i$ with $I(t) = \Sigma\alpha_i e^{-t/t_i}$, (b) $\Delta t = T_D - T_{DA}$ (in ns), (c) standard error of the mean, (d) total number of measured nuclei, (e) % FRET efficiency: $E = 1 - (T_{DA}/T_D)$, and (f) $p$ value of the difference between the donor lifetimes in the absence and presence of acceptor (Student's $t$ test). The statistical test used was two-sided. The lifetime measurements were carried out from two to three independent expression assays performed in $N.$ $benthamiana$ (leaf samples were taken between 36 and 48 h after infiltration with $A.$ $tumefaciens$).

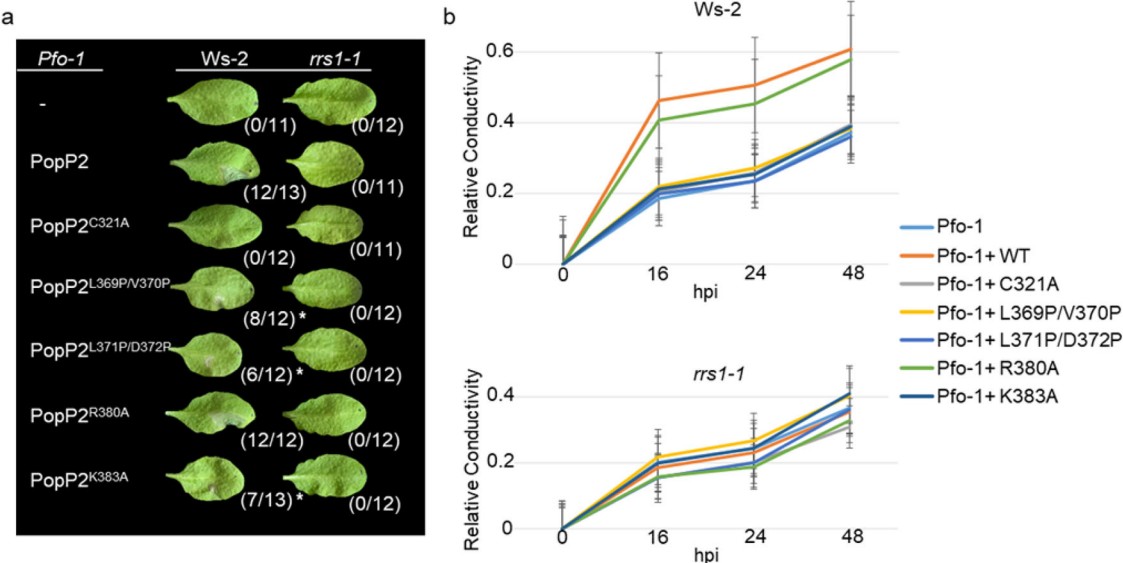

**Fig. 5 PopP2 K383, L369/V370, and L371/D372 residues are required for activation of RRS1-R-dependent immunity. a** $Pseudomonas$ $fluorescens$ ($Pf0$-$1$)-delivered PopP2 mutants K383A, L369P/V370P, and L371P/D372P trigger an attenuated RRS1-R-dependent cell death response in Ws-2 accession, compared to WT PopP2. Four-week-old Arabidopsis leaves were infiltrated with $Pf0$-$1$ strains delivering the indicated PopP2 proteins (WT PopP2, PopP2$^{C321A}$, PopP2$^{L369P/V370P}$, PopP2$^{L371P/D372P}$, PopP2$^{R380A}$, and PopP2$^{K383A}$). The photographs were taken at 72 h post infection (hpi), although the cell death response was already visible at 24 hpi. The numbers in brackets indicate the proportion of leaves developing a cell death response (HR hypersensitive response). The asterisk indicates a weaker HR observed compared to WT PopP2. This experiment was conducted three times with similar results. **b** Integrity of PopP2 L369/V370, L371/D372, and K383 residues is required for PopP2-triggered ion leakage in Ws-2. Data are mean ± SE ($n = 4$) of one representative experiment. Graphs indicate the ratio between released ions at each time point over total ion level measured at the end of the experiment upon sample boiling (relative conductivity). This experiment was conducted three times with similar results.

In Arabidopsis, PopP2 acetylation of RRS1-R WRKY domain activates RRS1-R-dependent immunity[13,14]. To further confirm the critical role of L369/V370, L371/D372, and K383 residues for PopP2 activity and substrate targeting, we investigated whether corresponding mutants were affected in the triggering of RRS1-R-dependent immunity. For this, we used a $Pseudomonas$ $fluorescens$ strain, $Pf0$-$1$, carrying a functional T3SS for delivery of PopP2 variants in Arabidopsis leaf cells[24]. As was previously described, WT PopP2 delivered by $Pf0$-$1$ triggered a cell death response already at 24 h post-infection (hpi) in the resistant Ws-2 accession carrying the $RPS4$ and $RRS1$-$R$ genes but not in $rrs1$-$1$ null mutant (Fig. 5a, photographs were taken at 3 days post-infiltration). By contrast, the catalytically inactive PopP2-C321A mutant was unable to trigger cell death in Ws-2, due to lack of

RRS1-R WRKY domain acetylation. Interestingly, the cell death response triggered by K383A, L369P/V370P, and L371P/D372P mutants in Ws-2 was affected compared to WT PopP2, suggesting the importance of these residues for PopP2 responsiveness (Fig. 5a). The amount of effector proteins delivered by $Pf0$-$1$ in plant cells was verified by immunoblot (Supplementary Fig. 9). To quantify the cell death response triggered by the different PopP2 variants, ion leakage assays were performed in Ws-2 and $rrs1$-$1$ plants. In agreement with the cell death macroscopic data described above, the single K383A mutation as well as the double L369P/V370P and L371P/D372P mutations induced less ion leakage than WT PopP2 or PopP2-R380A (Fig. 5b). Together, these data demonstrate the critical role of these PopP2 residues for substrate recognition/modification.

**A conserved mechanism likely adopted by YopJ family effectors to regulate substrate binding**. We then asked whether other YopJ effectors adopt similar strategy on the regulation of substrate binding. To answer this question, we compared all the solved structures of the YopJ family effectors. Besides PopP2 and HopZ1a, the crystal structure of AvrA, a YopJ family effector produced by animal pathogen *S. enterica*, was also reported recently in complex with InsP6 and CoA[25]. When superimposed, all the InsP6-bound structures contain a similar fold to that of the apo-PopP2, including the five-stranded β-sheet and its flanking helices, supporting the notion that the various YopJ family of acetyltransferases might be evolved from a common ancestor. In the PopP2-InsP6-CoA-RRS1$_{WRKY}$ structure, helix αD' plays a critical role in substrate recognition by interacting directly with RRS1$_{WRKY}$. The equivalent helix was also found in both HopZ1a and AvrA, presumably participating in substrate recognition[19,25]. Noticeably, all these helices are stabilized by a loop that is in a motif located at the C-terminus of the catalytic core, connecting the regulatory domain and a structurally conserved α-helix (αF' in PopP2-InsP6-CoA-RRS1$_{WRKY}$ complex, αG in HopZ1a-InsP6 complex, and αF in AvrA-InsP6-CoA complex) in the catalytic core (Fig. 6). In HopZ1a, this loop is linked to the regulatory domain through a β-strand, whereas in AvrA through a short α-helix. Therefore, the entire YopJ family effectors may adopt a common mechanism to regulate substrate binding, in which the motif at the C-terminus of the catalytic core is highly dynamic in the absence of InsP6, and InsP6 binding to the regulatory domain helps fix the conformation of this motif, thereby stabilizing the substrate recognition helix.

We also investigated whether other effectors in CE clan of proteases possess similar feature. On the phylogenetic tree, the YopJ family effectors are separated from the other CE clan of proteases (Supplementary Fig. 10), most of which are deubiquitinases[26]. Sequence alignment of the acetyltransferase domain of the YopJ family effectors and the catalytic domain of the deubiquitinases reveals that the existing regulatory domain is specific to the YopJ family members (Supplementary Fig. 11). These suggest that the way of substrate-binding regulation discovered here has been evolved independently among the YopJ family effectors.

## Discussion

InsP6, a small negatively charged metabolite that is present in the cytoplasm of eukaryote cells at micromolar concentrations[27], participates in numerous intracellular signaling pathways[28]. Extensive studies on the roles of InsP6 have outlined a basic paradigm for the action of InsP6, which modulates the activity of InsP6-binding proteins by altering their structure and/or surface charge topology[29]. The same strategy has been exploited by virulence proteins of some pathogens. For example, InsP6 activates the toxins TcdA and TcdB produced by *Clostridium difficile*[30,31] through the rotation of a small domain known as β-flap in their cysteine protein domains. In contrast, the conformational changes induced by InsP6 happen globally on YopJ family effectors, with the ligand binding and substrate binding regulated via two different mechanisms. Our previous studies have predicted the flexibility of the regulatory domain in the absence of InsP6. The apo-PopP2 structure reported here lends the first solid experimental evidence. The major contribution of this work is to link the InsP6-binding event to substrate-binding regulation through the discovery of a fold-switching motif. Therefore, the intradomain rearrangements observed here represents an important advance in our mechanistic understanding of the whole YopJ family effectors. However, it is still unclear why such a sophisticated activation process was developed in the course of evolution. One possibility is that the YopJ family effectors in their active state may be toxic to the bacteria producing them. To guard against such potentially deleterious enzymatic activity, bacterial pathogens must therefore ensure that these acetyltransferases are completely latent until they are delivered into host cells where the InsP6 co-factor makes the enzyme active and regulate binding to its substrates.

It is well known that globular proteins have a unique three-dimensional structure under physiological conditions. This notion has been challenged by the discoveries of fold-switching proteins. Fold-switching represent a key process in biology as it can potentially remodel the secondary structure of many proteins in response to cellular or environmental stimuli, generating active or inactive states[32,33]. Even though fold-switching proteins are believed to be widespread in nature[32], extensive protein structural studies have only identified a small number of them[34–37], especially those reshuffling secondary structures upon cellular stimuli, thus limiting our understanding of the principles behind their functions. In this study, we found that a fold switch happens in a small motif located between the regulatory domain and the substrate-binding helix of PopP2, serving as a transition system to relay the InsP6-binding signal from the regulatory domain to the substrate-binding helix. A random coil is revealed during the fold-switching process from a long α-helix to a β-hairpin. A similar process was also reported in the C-terminal domain of the transcription factor RfaH, in which an α-helical hairpin is refolded into a 5-stranded β-barrel through an unfolded state[38,39]. It would be worth investigating the universality of this process in

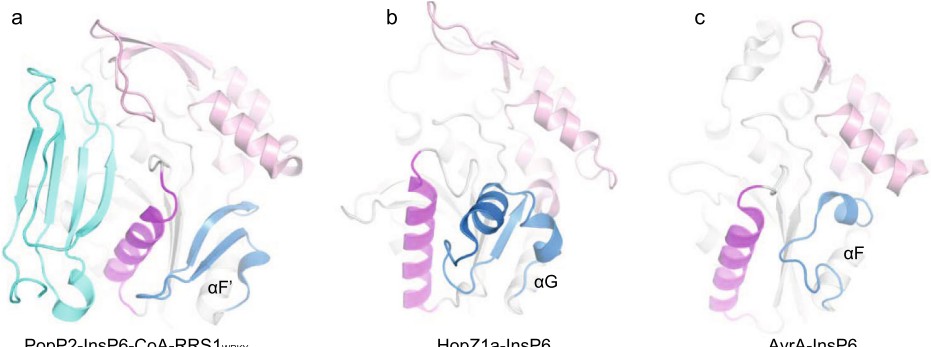

**Fig. 6 YopJ family effectors may adopt a conserved mechanism to regulate substrate binding by InsP6. a** Crystal structure of PopP2-InsP6-CoA-RRS1$_{WRKY}$ complex (PDB code: 5W3X). The regulatory domain is colored in pink; the fold-switching region is colored in blue; the substrate recognition helix is colored in purple; the RRS1$_{WRKY}$ domain is shown in cyan. Similar colors are used in **b**, **c**. **b** Crystal structure of HopZ1a-InsP6 complex (PDB code: 5KLP). **c** Crystal structure of AvrA-InsP6-CoA complex (PDB code: 6BE0).

related fold-switching proteins. The discovery of a fold-switch motif in PopP2 should greatly contribute to a better understanding of the molecular mechanisms regulating the activity of the other members of the YopJ family.

## Methods

**Expression and purification of recombinant PopP2 and RRS1$_{WRKY}$.** Briefly, genes encoding PopP2 (149–488) and RRS1$_{WRKY}$ were cloned into a modified pRSF-Duet vector, respectively, preceded by a His$_6$-SUMO tag. The fusion proteins were overexpressed in *E. coli* BL21(DE3) cells. Cells were induced with 0.4 mM isopropyl β-D-1-thiogalactopyranoside (IPTG) at an optical density at 600 nm of 0.8 and grown at 16 °C overnight. Expressed PopP2 proteins were first purified using a nickel column. The His6-SUMO tag was cleaved off by ULP1. The tagless protein was further purified using Phenyl HP hydrophobic interaction column (GE Healthcare) and the Superdex 200 size exclusion column pre-equilibrated with a buffer containing 25 mM Tris-HCl (pH 7.5), 250 mM NaCl and 2 mM dithiothreitol (DTT). The RRS1$_{WRKY}$ protein was purified through a Heparin column (GE Healthcare) followed by size exclusion chromatography on a Superdex 200 16/600 column pre-equilibrated with a buffer containing 25 mM Tris-HCl (pH 7.5), 100 mM NaCl, and 2 mM DTT[19].

Full-length PopP2 sequences (PopP2, PopP2$^{C321A}$, PopP2$^{L369P/V370P}$, PopP2$^{L371P/D372P}$, PopP2$^{R380A}$, and PopP2$^{K383A}$) flanked with attB1 and attB2 gateway sequences were generated by two-step PCR-based site-directed mutagenesis using PrimeStar HS DNA polymerase from Takara Bio Inc. (Otsu, Japan). PCR products were recombined using BP clonase in pENTR221 vector (Invitrogen). All DNA constructs were sequence-verified. The different sequences of PopP2 variants were introduced in the pETDuet-1-GWY gateway destination vector (as previously described in Le Roux et al.[14]) using LR clonase (Invitrogen). Single colonies (*E. coli* Rosetta (DE3) cells (Novagen)) transformed with relative pETDuet-1-gene destination vectors were grown on LB medium containing spectinomycin (50 µg mL$^{-1}$) and chloramphenicol (30 µg mL$^{-1}$) at 37 °C to an OD$_{600nm}$ of 0.6 and induced with 250 µM IPTG for 4 h at 28 °C under shaking. Pelleted cells were concentrated 10 times in phosphate-buffered saline (pH 8.0) supplemented with 1 mM phenylmethylsulfonyl fluoride (Sigma Aldrich), 10 mM sodium butyrate, 0.1% Triton, 1 mM EDTA, and 2.5 mM DTT and lysed for 5 min on ice using BugBuster 1× reagent (Millipore) supplemented with Lysonase according to the manufacturer's instructions. After centrifugation, supernatants were recovered and denaturated in laemmli buffer (2× final concentration) for 5 min at 95 °C. Proteins samples were separated by sodium dodecyl sulfate-polyacrylamide gel electrophoresis (SDS-PAGE) gel (Bio-Rad) and subjected to immunoblot analysis. Transferred proteins were visualized by Ponceau S red staining.

**Crystallization and structure determination.** Crystallization was carried out using hanging-drop diffusion method. The concentration of PopP2 was about 40 mg mL$^{-1}$. Equal volumes of PopP2 and the precipitant solution (0.1 M HEPES (pH 7.5), 50 mM MgCl$_2$, 30% PEG55O MME) were mixed. Crystals appeared within 2 days at 4 °C and grew to full size in a week. Crystals were cryo-protected using the precipitant solution supplemented with glycerol to a final concentration of 20% (v/v) before flash frozen in liquid nitrogen. Datasets were collected on the beamline BL19U1 at the Shanghai Synchrotron Radiation Facility. The data were indexed, integrated, and scaled by the HKL2000 package[40]. The structural solution was obtained by molecular replacement using PHASER[41] in Phenix[42] and the core region of InsP6-bound PopP2 structure (residues 149–370) as the search model. Iterative rounds of model building in COOT[43] and refinement in Phenix were carried out. Data collection and refinement statistics are summarized in Supplementary Table 1.

**BLI assays.** The binding affinities between PopP2 variants and WRKY in the presence of InsP6 were measured by BLI assay on an OKTET K2 system (ForteBio Inc., Menlo Park, CA, USA). All the PopP2 variants contained an AVI-tag on the N-terminus, which was biotinylated by the bacterial biotin ligase BirA in a buffer containing 25 mM Tris-HCl (pH 8.0), 250 mM NaCl, and 2 mM DTT. The modified PopP2 proteins were then immobilized onto capture streptavidin biosensors and balanced with buffer E (25 mM Tris-HCl (pH 8.0), 250 mM NaCl). The biosensors were then exposed to different concentrations of RRS1$_{WRKY}$, followed by dissociation in buffer E. Binding affinities ($K_D$) were calculated using the DataAnalysisHT software.

**In vitro acetylation assays.** In vitro acetylation assays were carried out using an Ac-Lys antibody (Santa Cruz Biotechnology; dilution 1:2000). A reaction mixture of 20 µL for each PopP2 variant contains 0.5 µg PopP2, 40 µg RRS1$_{WRKY}$, 125 µM AcCoA, and 2 mM InsP6 in the reaction buffer (25 mM Tris-HCl (pH 8.0), 250 mM NaCl, and 5% glycerol). The reaction was performed at room temperature for 1 h and then 5 µL of each sample was subjected to 12% SDS-PAGE for western blot. Another SDS-PAGE gel with the same amount of sample loaded was stained with Coomassie Blue as loading control.

**DNA-binding assay.** The acetyltransferase activity of PopP2 variants were monitored using RRS1$_{WRKY}$ domain as substrate. Four micrograms of WT PopP2 or mutant was incubated with 40 µg of RRS1$_{WRKY}$ and 10 µg of W-box dsDNA (upper strand, 5′-CGCCTTTGACCAGCGC-3′) in 25-µL reactions (25 mM Tris-HCl, pH 8.0, 5% glycerol, 250 mM NaCl, 1 mM DTT, 1 mM AcCoA, and 2 mM InsP6) on ice for 25 min. The samples were loaded onto 12% non-denaturing polyacrylamide gel and electrophoresed at 0 °C in 0.5× TBE buffer. The gel was stained with ethidium bromide and visualized under ultraviolet light. Original results are provided in the Source data.

**Enzyme conformation monitoring using Raman signals.** To detect the enzyme conformation, the fabricated Au@Ag NPs were prepared by sodium citrate reduction of HAuCl$_4$[44] and used as substrates for the Raman signal enhancement. All enzymatic stock solution was diluted to 1 mg mL$^{-1}$ and then was mixed with the appropriate amount of Au@Ag NPs. The mixture was dropped on a slide and the Raman spectra were recorded using HORIBA LabRAM HR Evolution Raman spectrometer (HORIBA, France) equipped with 532 nm laser and ×50 objective.

**Plant material and growth conditions.** *Arabidopsis thaliana* accession Ws-2 and *rrs1-1* null mutant (Ws-2 genetic background) plants were grown in Jiffy pots in short day conditions (22 °C, 60% relative humidity, 125 µE M$^{-2}$ s$^{-1}$ fluorescent illumination, 8 h light/16 h dark cycles).

**P. fluorescens-mediated delivery of PopP2 variants.** For Pf0-1-mediated delivery, full-length PopP2 variants (PopP2, PopP2$^{C321A}$, PopP2$^{L369P/V370P}$, PopP2$^{L371P/D372P}$, PopP2$^{R380A}$, and PopP2$^{K383A}$) were recombined in the pBBR-AvrRps4prom-GWY-3HA gateway destination vector using LR clonase (Invitrogen)[14]. pBBR clones were introduced in Pf0-1 by triparental mating using the pRK2013 helper strain. Transformed Pfo-1 cells were selected on King's B (KB) agar supplemented with 6 mM MgS04 and with appropriate antibiotics (tetracycline 5 µg mL$^{-1}$, chloramphenicol 30 µg mL$^{-1}$, and gentamicin 15 µg mL$^{-1}$).

**Plant pathology experiments.** For PopP2-triggered cell death assays, *P. fluorescens* (Pf0-1) cells were grown overnight at 28 °C on KB agar supplemented with appropriate antibiotics and harvested in 10 mM MgCl$_2$. The final concentration of Pf0-1 cell suspensions was adjusted to OD$_{600}$ = 0.2. Leaves of 4-week-old Arabidopsis plants (Ws-2 and rrs1-1 null mutant) were hand-infiltrated on the abaxial surface using a blunt-end syringe. Macroscopic symptoms were observed between 24 and 72 hpi and photographed at 72 hpi. For ion leakage assays (conductivity measurements) with Pf0-1 cells, for each measured sample, 4 leaf disks (4 mm$^2$) were sampled at 3 hpi, washed in 1800 µL of nano-pure water for 60 min (with gentle shaking at room temperature) and transferred to fresh 800 µL of nano-pure water (0 hpi sample). Ion leakage measurements were performed at time points indicated using a conductivity meter (Horiba LAQUAtwin B-771).

**Plasmid constructs used for in planta acetylation assay and time-correlated single photon counting (TSPC)-FLIM measurements.** All full-length PopP2 variants were expressed as a C-terminal fusion either with a triple HA tag or CFP after recombination in the pAM-PAT-35S-GWY-3HA and pAM-PAT-35S-GWY-CFP destination vectors, respectively. The pB7FWG2 vector used for expression of the C-terminal portion of RRS1-R fused with the enhanced GFP variant (RRS1-R$_{Cterm}$-eGFP) was previously described (Le Roux et al.[14]). For FRET-FLIM assay, the RRS1-R$_{Cterm}$-YFPv construct used was obtained after LR reaction between pENTR207-RRS1-RCterm and pAM-PAT-35S-GWY-YFPv[23]. All pAM-PAT-derived plasmids were introduced by electroporation in *A. tumefaciens* GV3103. The recombined pB7FWG2 plasmid used for expression of RRS1-R$_{Cterm}$-eGFP fusion was transformed in *A. tumefaciens* GV3101. Primers used to generate different constructs are provided in Supplementary Table 3.

**Transient assays in N. benthamiana.** For *Agrobacterium*-mediated *N. benthamiana* leaf transformation, the relevant GV3103 and GV3101 strains were grown in Luria-Bertani liquid medium with appropriate antibiotics (25 mg mL$^{-1}$ gentamicin and 25 mg mL$^{-1}$ carbenicillin for pAM-PAT-derived plasmids; 25 mg mL$^{-1}$ gentamicin and 50 mg mL$^{-1}$ spectinomycin for pB7FWG2 plasmids) for 24 h at 28 °C. Bacteria were harvested and resuspended in infiltration medium (10 mM MES pH5.6, 10 mM MgCl$_2$, 150 µM acetosyringone). Suspensions were then adjusted to OD$_{600}$ = 0.25 and incubated for 2 h at room temperature before leaf infiltration. For acetylation assay, the ratio was 1:1. For FRET-FLIM assay, bacterial suspension for donor and acceptor constructs were mixed in a 1:4 ratio (final OD$_{600}$ = 0.1 and 0.4, respectively). The infiltrated plants were incubated between 36 and 48 h in growth chambers under controlled conditions. Original results are provided in the Source data.

**Protein extraction and immunoblotting.** Arabidopsis protein samples were prepared from rrs1-1 null mutant, 7 h after infiltration with Pf0-1 cells. For each sample, eight leaf discs (7 mm$^2$) were ground in liquid nitrogen and total proteins were extracted in 1 mL of ice-cold IP buffer (50 mM Tris-HCl pH7.5, 150 mM NaCl, 10 mM EDTA, 2 mM DTT, 1× Plant protease inhibitor cocktail (SIGMA),

and 0.1% Triton). Lysates were centrifuged for 5 min at $20{,}000 \times g$ at 4 °C. The supernatants were filtered through miracloth mesh (Millipore). For crude extracts, 50 μL of supernatant was mixed with 50 μL of 4× Laemmli buffer and denatured for 3 min at 95 °C. For immunoprecipitation, supernatants were incubated with 10 μL of anti-HA magnetic beads (Pierce) during 1 h at 4 °C under gentle shaking. Beads were washed one time in 800 μL IP buffer for 5 min at 4 °C. Immunoprecipitated proteins were denatured in 40 μL of 4× Laemmli buffer for 3 min at 95 °C. For immunoprecipitation of fluorescent-tagged proteins, protein samples from *N. benthamiana* leaves (4 discs of 8 mm diameter) were homogenized in 1 mL of ice cold IP Buffer. The extract was centrifuged at $13{,}000 \times g$ for 2 min at 4 °C. Ten microliters of GFP-binding protein affinity matrix (Chromotek) was added to the supernatant and rotated at 4 °C for 60 min. Samples were centrifuged for 3 min at $1000 \times g$. Beads were washed three times with IP buffer and subsequently boiled in 2× Laemmli buffer for 3 min at 95 °C. Crude extracts and immunoprecipitated protein were analyzed as indicated before.

Samples were separated by SDS-PAGE and transferred on nitrocellulose membrane for proteins visualization (Ponceau S red staining). Membranes were blocked in a 2% milk-TBS-T solution for 1 h before immunoblotting. The following primary antibodies were used: anti-Acetylated Lysine (Ac-K-103, Cell Signaling Technology; dilution 1:2000), anti-HA-HRP (3F10; Roche; dilution 1:5000), anti-GFP (mouse monoclonal; Roche; dilution 1:3000), and anti-His6-HRP (mouse monoclonal; Roche; dilution 1:50,000). The appropriate horseradish peroxidase (HRP)-conjugated secondary antibodies were applied (goat anti-mouse IgG-HRP (Bio-Rad, dilution 1:10,000) for detection of anti-GFP; goat anti-mouse IgG2a-HRP (Bio-Rad, dilution 1:5000) for detection of Ac-K103). Proteins were detected using Clarity Reagent (BioRad). Original results are provided in the Source data.

**TSPC-FLIM data acquisition**. FLIM was performed on Leica TCS SP8 SMD, which consists of an inverted LEICA DMi8 microscope equipped with a TCSPC system from PicoQuant. The excitation of the CFP donor at 440 nm was carried out by a picosecond pulsed diode laser at a repetition rate of 40 MHz, through an oil immersion objective (×63, N.A. 1.4). The emitted light was detected by a Leica HyD detector in the 450–500 nm emission range. Images were acquired with acquisition photons of up to 1500 per pixel. From the fluorescence lifetime images, the decay curves were calculated per pixel and fitted (by Poissonian maximum likelihood estimation) with a tri-exponential decay model using the SymphoTime 64 software (PicoQuant, Germany).

**MD simulations**. MD simulations were performed with GROMACS-2018.4 using the Amber99sb-ildn force field. The force field parameter for $InsP_6$ were obtained using the ACPYPE script. The apo- and InsP6-bound PopP2 were prepared using the Protein Preparation Wizard in the Schrödinger software. The missing loops and domain in the protein structure were modeled with Prime in the Schrödinger software. The systems were solvated with TIP3P waters, with the box size as 7.8 nm × 7.8 nm × 7.8 nm. In all, 150 mM NaCl were added to the systems to mimic the biological environments. Then the systems were minimized with a 50,000-step energy minimization using the steepest decent algorithm. The systems were subjected to temperature equilibrating in the NVT ensemble at 300 K for 200 ps, after that density equilibrating in the NPT ensemble at 300 K and 1 atm for 400 ps. All the heavy atoms were constrained using a harmonic restraint with the force constant set to 1000 kJ mol$^{-1}$ nm$^{-2}$ in the equilibrating steps. The production runs lasted for 500 ns and 3 replicas were used to ensure reproducibility. All the production runs were carried out under NPT ensemble.

**Molecular mechanics/generalized Born surface area**. AMBER18 was used to obtain the trajectories as well as the MM/GBSA calculations. The force field parameters for zinc-binding domain were obtained using the MBCP.py script. The simulations lasted for 20 ns and the final 5 ns were used for the MM/GBSA calculations.

**Metadynamics**. Metadynamics simulations were conducted using the GROMACS-2018.4 patched with plumed-2.5.1. The well-tempered variant of metadynamics were used. The metadynamics simulation lasted for 800 ns. Path-CVs were used in metadynamics. A reference path is defined by a set of conformations that we obtained from targeted MD. For the targeted MD, the bias is added on a RMSD coordinate. The RMSD is the backbone atoms of the region 350–375 (represents the fold-switch motif) with respect to it in the apo state, using the CA atoms in the region 10–199 to do the alignment. The targeted MD simulation was performed using the GROMACS-2018.4 and PLUMED-2.5.1. After the targeted MD, we selected 44 conformations in the targeted MD trajectory as the metric that was used in the path-CVs. The progress along this path (s-path) is defined according to the following equation:

$$S(X) = \frac{\sum_{i=1}^{N} i \exp^{-\lambda|X-X_i|}}{\sum_{i=1}^{N} \exp^{-\lambda|X-X_i|}} \tag{1}$$

where $X$ represents a conformation, $N$ is the number of conformations that defined the path, $\lambda$ is a smoothing parameter, $i$ represents the $i$th conformation, and $|X - X_i|$ is the mean-square deviation of a subset of atoms in conformation $X$ to the $i$th conformation $X_i$.

The sampling is further enhanced by using z-path, which is defined with the following equation:

$$Z(X) = -\frac{1}{\lambda} \log \left( \sum_{i=1}^{N} \exp^{-\lambda|X-X_i|} \right) \tag{2}$$

which represents the deviation away from the structures on the reference path. An example trajectory of the metadynamics simulation is shown in Supplementary Movie 4.

**Reporting summary**. Further information on research design is available in the Nature Research Reporting Summary linked to this article.

## Data availability

The atomic coordinates and structural factors generated in this study have been deposited into the Protein Data Bank under the accession code 7F3N. Other data are available from the corresponding authors upon reasonable request. Source data are provided with this paper.

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

## Acknowledgements

We sincerely thank Dr. Jikui Song for revising this manuscript. This work was supported by funds from Natural Science Foundation of China (31800638, 3217110027) and Natural Science Foundation of Guangdong Province (2018A030313003) to Z.-M.Z. The work of S.Y. was supported by funding from Chinese Academy of Sciences, the Shenzhen Institutes of Advanced Technology, CAS, Shenzhen government (JCYJ20200109114818703) as well as that from Guangdong province (2019QN01Y306). The work of M.E. and L.D. was supported by a research grant from the Agence Nationale pour la Recherche (ANR-18-CE20-0015). This work was supported by the French Laboratory of Excellence project "TULIP" (ANR-10-LABX-41; ANR-11-IDEX-0002-02) and the "École Universitaire de Recherche (EUR)" TULIP-GS (ANR-18-EURE-0019). We thank the staff from BL17B/BL18U1/BL19U1/BL19U2/BL01B beamline of National Facility for Protein Science in Shanghai (NFPS) at Shanghai Synchrotron Radiation Facility for assistance during data collection.

## Author contributions

Y.X. and C.Z. crystallized PopP2 and determined the structure. G.L., Y.X., Y. W. and L.Z. performed the BLI assay and in vitro enzymatic assay. M.E. and L.D. performed all the in planta-related assays. C.P. and L.D. performed FLIM-FRET assays. R.Z. and S.Y. performed computational study. Q.W., H.Z. and P.S. performed Raman experiment. K.D. and Z.-M.Z. conceived and oversaw the project. L.D., S.Y. and Z.-M.Z. wrote the manuscript.

## Competing interests

The authors declare no competing interests.
