## [Peer Review File · Nature Communications]

Secondary-structure switch regulates the substrate binding of a YopJ family acetyltransferaseREVIEWER COMMENTS

Reviewer #1 (Remarks to the Author):

Xia et al. report a new 3D structure of a Yersinia outer protein J (PopP2) in apo form, determined using X-ray crystallography. By comparing with the previously determined structure of PopP2 in complex with inositol hexaphosphate (InsP6), the authors reveal interesting structural transformation of a 25-amino acid region (residues 351-375) that (upon the binding InsP6) switches from an alpha helix to a beta hairpin. This work thus adds to the growing body of work on how fold switching in proteins, i.e., changes in secondary and tertiary structures, is used by proteins to regulate function.

Additionally, this work reveals two other minor structural transformations in PopP6 upon InsP6 binding: the formation of an alpha helix (alphaD) in a flexible loop region and a partial melting of helix alphaG. Hence, this protein exhibits three different types of structural transformations triggered by a single binding event: fold switching, disorder-to-order and order-to-disorder. This rather unique feature could be further emphasized in the text. The structural transitions of PopP6 are investigated using biochemical and computational analyses.

I am convinced that this work will be of significant interest to researcher in the field of fold switching and more generally in protein biophysics. However, I have some comments and concerns, which are listed below. These should be addressed before the manuscript can be accepted for publication.

Major:

Principle Component Analysis (PCA) and metadynamics simulations are the two main computational analyses carried out. These are reasonable choices and have the potential to clarify important aspects of the fold switch. Unfortunately, both methods used are described only in brief. Additional details of these calculations should be provided either in Methods or in Supplementary Information. In the metadynamics, what are the collective variables used? Figure 3 indicates the path collective variables are “s-path” and “z-path” but no explanation is given for the reader in either the text or figure legend. To achieve an initial transition a targeted MD simulation was carried out (line 439). How was this simulation carried out? What metric was used to define the PCVs? What does a typical metadynamics simulation trajectory look like? An example trajectory should be given. Regarding the PCA: while the preparation of the initial structures are describe in detail, basic information is missing on the production runs. For example, in which ensemble was the simulations carried out?

In my opinion, the Discussion section is a little general and could be strengthened. It focuses on the mechanism of PopP6 action and on how PopP6 relates to other fold switching proteins. This focus is appropriate. However, much of the Discussion has an introductory nature (could some parts be moved in Introduction?). Connections to previous work is limited to two sentences at the end of each of the two paragraphs and are rather general.

Minor comments:

1) Figures 1 and 2 need some work. It should be noted in Figure 1 legend that the regulatory domain 377-447 is absent from the apo state. In a), secondary structure elements for both states should be included. This would greatly help to illustrate the structural changes that take place. Much of Figure 2 appears unnecessarily shaded. For example, features highlighted by boxes should not be shaded at all (it appears counterproductive to me). Colors in Figure 2 appear to be the same as in Figure 1, but mean different things. This should be avoided.

2) The term “fold-switching motif” is used for the first time on line 160 and thereafter repeated throughout the text. For clarity, at first mention, specify which residues make up this motif.

3) Why does the peak at 1657cm⁻¹ in the Raman spectrum disappear completely after incubation with InsP6 when much of the alpha helical structure remains in the structure?

4) On line 176, a region between the two basins of attraction has been designated “transition state”. I think this is too strong. No evidence is presented that the selected region satisfies suitable criteria for a transition state (e.g., that this point kinetically partitions the two basins).

5) On line 184 “ To test this possibility, we firstly performed a 3x500 ns all-atom molecular dynamics (MD) simulations for apo and InsP6-bound PopP2”. Presumably, it is meant 3x500 ns for *each of* the two structures? Please specify.

6) The RMSD values reported on line 189 should be clarified. Are they the values at the end of the simulations? Averages taken over part of simulations?

7) Line 193 “calculation of binding energy”. Specify the method used.

8) The conclusion that the fold switches goes via a relatively disordered intermediate conformational ensemble is interesting. The behavior is similar to that found for the C-terminal domain of the RfaH protein (see, for example, Biopolymers. 2021;e23420 or J. Phys. Chem. B 2014, 118, 5101). This can be worth pointing out.

9) On line 420, specify the Schrödinger software (they have many products).

10) Regarding MM/GBSA calculations, it would be nice if the authors could present a dissection of the binding energy in terms of enthalpy and entropy components, as is often done for this kind of binding affinity calculations. Specifically, the PCA analysis suggests a much more flexible RRS1WRKY binding region in the apo state than the InsP6 bound state. Can this be confirmed by different Delta S components of the binding energy between the two states?

Reviewer #2 (Remarks to the Author):

The authors present a paper describing the mechanism of allosteric activation by inositol hexaphosphate (InsP6) of a YopJ T3SS effector family member (PopP2) from the plant pathogen *Ralstonia solanacearum*. They present for the first time an Apo structure of PopP2 (not bound to InsP6). Comparison of the Apo structure to that of other PopP2-InsP6 Complexes revealed binding in a remote site to dictate a conformation switch in the catalytic site that also stabilized the substrate recognition helix. The authors support the structure observations with biochemical and computational data and provide a comparison to other YopJ family members with solved structures. Although the structure (with caveats of its instability and poor structure validation metrics) depicts an interesting fold switch, the paper describing previous PopP2 structures (PMID: 28737762) already reported similar disordered segments in MD simulations in the absence of InsP6. More importantly, the authors do not discuss how the switch relates to the enzyme mechanism (and some of their experiments were designed without its' consideration). Instead, the authors provide a somewhat lengthy discussion of fold change, listing several random examples where switching occurs without appropriate reference to the rich history of structural plasticity that had been described for decades.

The description of the active site in the introduction is confusing, which residues contribute to the acetyltransferase activity of PopP2?

Figure 2 is difficult to see the conformation changes. Perhaps using different colors for the parts of the structure that move? For example, the light blue apo structure could have blue for the parts described in the text as having a conformation change and the InsP6 could have red.

In Figure3B, the minima look the same to me. Is the difference between the apo and InsP6 bound state significant? Why is it “as expected” that the global minima would be for the apo state? MD simulations in PMID 28737762 suggest the same region from 350-370 has increased rmsf. Can the global energy minima for the helix in the apo state be explained in light of the published rmsf for the

same region in the apo state? I can not tell from the PCA arrows in figure S3 if the MD simulations recapitulate the published data.

Figure 4A looked at the binding of only two InsP6 residues (R380 and K383). What about K453, which trades a hydrogen bond with a neighboring Ser in the Apo state with a hydrogen bond to InsP6 when it binds? This residue seems relevant to the mechanism being described by the authors (instead of reporting K383 is the key residue) and should not be excluded. The BLI binding curves are performed in a background active site C321A mutation, which is not stated in the text. Why? Especially considering the reported ping pong mechanism of acetyltransferase activity where AcCoA binds first and acetylates this Cys prior to RRS1 binding. The L369P/V370P and L371P/D372P seem quite disruptive to both the helix and hairpin conformations of the switch region. Are these mutations stable? Can they bind InsP6 or AcCoA? The crude acetylation assay in figure 4c does not measure acetylation, it measures binding to DNA. The statement “unacetylated RRS1 proteins do not form complex with W box” is not shown anywhere in the paper.

Figure 5 has typos in the figure legend (wilt for wild) and experiments in 5a should be performed in triplicate for acceptable rigor.

Figure S3 c and d have several lines of colors that are not defined in the figure legend.

Although the name might be relatively new and coined in the referenced review, the concept of “fold switching” is not recent. The term “switch” was described in 1990 for changes in active/inactive ras protein structures. Similar studies describe chameleon sequences (2015) in homologs that reflect structural plasticity. Also reviews since 2002 describe the plasticity of protein folds PMID: 12127461.

Line 123 well reserved?

Line 127 ahead of

Line 145 change lack of to lacking

Line 154 the regulatory

Line 169 simulation, which

Line 175 with what was observed

Line 169-170 path collective variables

Line 190 switch spelling

Line 286 to from?

Line 289 represent

Reviewer #3 (Remarks to the Author):

This article by Yao Xia et al. addresses the molecular mechanism of inositol hexaphosphate (Insp6)-mediated allosteric regulation of YopJ family acetyltransferases. The manuscript highlights the previously undescribed fold-switching mechanism that regulates the substrate binding of YopJ family acetyltransferases by comparing the crystal structure of PopP2, a YopJ family type III secreted effector in apo- and Insp6-bound state. Furthermore, using biochemical and computational analysis the authors further demonstrated the chain reaction of conformational changes induced by the interaction of Insp6 with PopP2. The manuscript is well written and provides interesting findings underlying allosteric regulation of YopJ family acetyltransferases. However, there are few weaknesses.

Specific comments:

The authors demonstrated how Insp6 regulates substrate binding of PopP2 through fold switching motif. To address this, the authors used various biochemical and biophysical approaches to investigate the role of residues in the Insp6-binding pocket and fold-switching motif in mediating the interaction of PopP2 with RRS1WRKY. Using bio-layer interferometry (BLI) assay and in vitro acetylation assay, the author highlighted the significance of K383A (in Insp6 binding pocket) and

L369-D372 (in fold-switching motif), in regulating the interaction between PopP2 and RRS1WRKY. However, any evidence of the effect of these mutations on the overall conformation of PopP2 is missing.

Figure 4(c): It would be good to add additional panel (either in main figure or as a supplementary image) displaying the acetylated state of RRS1WRKY by PopP2 wild type and variants in the presence of InsP6, AcCoA.

Figure 5(b): The PopP2 mutants mentioned in the figure legend for the ion-leakage assays seem incorrect. It should be L369P/V370P and 371P/D372P instead of L369A/V370A and 371A/D372A.

Minor comments:

Line 588 - wilt > wild

Line 190 - swithc > switch

Line 169-170 - path collective various > path collective variables

Figure S2: Figure legend – color of InsP6-bound PopP2 should be pink not red.

Reviewer #4 (Remarks to the Author):

This manuscript titled “Secondary-structure switch regulates the substrate binding of a YopJ family acetyltransferase” describes structural studies of inactive (apo-state) acetyltransferase type III secreted bacterial effector PopP2 and the conformational changes undergone by binding critical host-derived co-factor InsP6. The work uses crystal structure of inactive PopP2 (in the absence of InsP6) coupled to the previously published crystal structure of InsP6-bound PopP2 to ascertain a fold shift in PopP2 that is likely representative of other effectors from this large family as described in the manuscript. The authors also use Raman spectra, metadynamics simulation, and bio-layer interferometry and in vitro acetylation to demonstrate key residues required for this process. Finally, authors show, using classical biological assays, that such residues determine PopP2’s ability to affect its host target (the WRKY domain of RRS1 as a proxy for other true targets: WRKY TFs) in its ability to bind DNA. The findings of the study clearly demonstrates a long-suspected role of InsP6 in making PopP2 (and other acetyltransferase effectors) competent for substrate binding, particularly through a novel fold-switching modification. The study is of some interest to the broader scientific community and findings may be applicable for future development of drugs that could target the co-factor binding ability of this effector family from plant and animal pathogens. I feel that the scope of this manuscript does indeed suit this journal, but several issues with this manuscript will need addressing first.

The following major issues should be addressed prior to acceptance for publication:

1. Inclusion of a phylogenetic tree of effectors from this family (including PopP2, AvrA, YopJ and HopZ1a) will be important for understanding the conclusions reached in this manuscript. This is particularly interesting if this was presented specifically for the ‘regulatory domain – aa377-447’ or the ‘acetyltransferase domain – aa149-488’ described in Figure 1. I also recommend adding corresponding regions from several non-acetyltransferase effectors from the CPD family to delineate where the broader conclusions about the YopJ family stand with regard to other CPD effectors.
2. L211-212: Authors suggest that the “L369P/V370P and L371P/D372P mutations completely abolished acetyltransferase activity (Fig 4c)” but the figure panel only demonstrates DNA binding by the RRS1 WRKY DNA-binding domain and not acetylation status of this domain. As such, this would be ideal using an anti-Acetyl Lysine antibody (α -AcK) for a western blot. This assay has not been used in this manuscript and should be included to demonstrate that acetylation is in fact affected leading to the predicted antagonism of phosphorylation required for immunity (Guo et al. 2020 CHM, DOI: 10.1016/j.chom.2020.03.008).
3. L179-214: Authors demonstrate using molecular dynamics and BLI assays to demonstrate that RRS1-WRKY binding by PopP2 is affected by the InsP6-binding triggered fold-switch in PopP2. However, they have not shown this via a co-immunoprecipitation (co-IP) assay. Is binding in a co-IP assay affected? Is there a reduction in acetylation (assessed by α -AcK blot)? This data would be both interesting and highly relevant to this manuscript.
4. L260-262: Authors report that the B-factors for loops associated with the substrate-binding domain

are higher than for the catalytic core. These values (comparison between both) are not reported and not demonstrated in a scale bar in Supplementary Fig. S6. Furthermore, is the conclusion about the YopJ family effectors (L261-264) correct regarding flexibility of these interfaces in the absence of InsP6 when they have not been measured under these conditions, but rather have apparently only been calculated for InsP6-bound structures?

The manuscript is generally well written, apart from a number of grammatical and spelling errors outlined below:

L27: “employed in” is strange terminology. Perhaps “deployed through” is better.

L39: “form target” > “form a target”

L51: “to rapid” > “to the rapid”

L54: “family which are” > “family, whose members are”

L62: “protease” > “proteases”

L63: “amounting” > “mounting”

L67: “targets” > “target”

L70: “to suppressed” > “to a suppressed”

L77: “root infecting” > “root-infecting”

L82-83: This sentence is very vague. What “about the modification sites” is unclear?

L84: “between YopJ” > “between the YopJ”

L120: “unable to build, due” > “unable to be built due”

L123: “well reserved” – do you mean “well-conserved”?

L145: “constrained, lack” > “constrained, with lack”

L149: “in recognition” > “in the recognition” – should ‘recognition’ instead be ‘interaction’?

L154: “regulatory domain” > “the regulatory domain”

L158: “and regulatory” > “and the regulatory”

L164: “in amide” > “in the amide”

L165: “ α -helix” > “the α -helix”

L170: “various” > “variables”

L171: “ α -helix into β -strand” > “an α -helix into a β -strand”

L175: “what observed” > “what is observed”

L176: “in to” > “into”

L179: “Insp6” > “InsP6”; “substrate binding” > “substrate-binding” (the latter change needs to be made throughout the manuscript, not listed here further)

L180: “InsP6 binding” > “InsP6-binding”

L184: “performed a 3x500” > “performed 3x500”

L187: “that the apo PopP2” > “that apo PopP2”

L190: “switch” > “switch”

L191-192: “leading to relatively smaller and deeper pocket which” > “leading to a relatively smaller and deeper pocket, which”; “facilitate a stronger” > “facilitate stronger”

L195: “-85.3 kcal/mol vs -59.7 kcal/mol” > “-85.3 kcal/mol vs -59.7 kcal/mol, respectively”

L196: “assay” > “assays”

L197: “assay” > “assays”

L204: “between InsP6 binding pocket and substrate” > “between the InsP6 binding pocket and the substrate”

L210: “K383A mutation” > “the K383A mutation”

L215: “The PopP2-triggered” > “PopP2-triggered”

L217: “RRS1-R WRKY” > “the RRS1-R WRKY”

L228: “cell death response” > “the cell death response”

L235: “single K383A mutation” > “the single K383A mutation”

L247: Remove “members”

L286: Unclear grammatical error, please review

L294: Unclear grammatical error, please review

L297-298: “believed to widespread” > “believed to be widespread”

We thank all the reviewers for their critical comments on our work. We tried our best to address all their concerns and revised the manuscript accordingly. Please find our point-to-point response to each of the reviewers' comments below.

REVIEWER COMMENTS

Reviewer #1 (Remarks to the Author):

I am convinced that this work will be of significant interest to researcher in the field of fold switching and more generally in protein biophysics. However, I have some comments and concerns, which are listed below. These should be addressed before the manuscript can be accepted for publication.

Response: We are grateful to the reviewer for pointing out the significance of this work. Our responses to the reviewer's comments are listed below.

Major:

Principle Component Analysis (PCA) and metadynamics simulations are the two main computational analyses carried out. These are reasonable choices and have the potential to clarify important aspects of the fold switch. Unfortunately, both methods used are described only in brief. Additional details of these calculations should be provided either in Methods or in Supplementary Information.

Response: Following the reviewer's suggestion, we have added details of these calculations in the Method part of the revised manuscript.

In the metadynamics, what are the collective variables used?

Response: In the metadynamics simulation, we used path collective variables (path CV) to enhance sampling. A reference path is defined by a set of conformations that we obtained from targeted MD. The progress along this path (s-path) is defined according to the following equation:

$$S(X) = \frac{\sum_{i=1}^N i \exp^{-\lambda|X-X_i|}}{\sum_{i=1}^N \exp^{-\lambda|X-X_i|}},$$

where X represents a conformation, N is the number of conformations that defined the path, λ is a smoothing parameter, i represents the i -th conformation, and

$|X - X_i|$ is the mean-square deviation of a subset of atoms in conformation X to the i -th conformation X_i .

The sampling is further enhanced by using z-path, which is defined with the following equation:

$$Z(X) = -\frac{1}{\lambda} \log \left(\sum_{i=1}^N \exp^{-\lambda|X-X_i|} \right),$$

which represents the deviation away from the structures on the reference path.

All the biased and unbiased simulations were carried out under NPT ensemble.

Figure 3 indicates the path collective variables are “s-path” and “z-path” but no explanation is given for the reader in either the text or figure legend.

Response: “s-path” means progress along the path, “z-path” means deviation along the path. We added these explanations in the figure legend in the revised manuscript.

To achieve an initial transition a targeted MD simulation was carried out (line 439). How was this simulation carried out?

Response: For the targeted MD, the bias is added on a RMSD coordinate. The RMSD is the backbone atoms of the region 351-368 (represents the fold-switch motif) with respect to it in the apo state, using the CA atoms in the region 155-344 to do the alignment. The targeted MD simulation was performed using the GROMACS-2018.4 and PLUMED-2.5.1.

What metric was used to define the PCVs?

Response: After the targeted MD, we selected 44 conformations in the targeted MD trajectory as the metric that used in the path-CV.

What does a typical metadynamics simulation trajectory look like? An example trajectory should be given.

Response: An example trajectory is shown in Movie 4.

Regarding the PCA: while the preparations of the initial structures are describe in detail, basic information is missing on the production runs. For example, in which ensemble was the simulations carried out?

Response: All the biased and unbiased simulations were carried out under NPT ensemble.

In my opinion, the Discussion section is a little general and could be strengthened. It focuses on the mechanism of PopP6 action and on how PopP6 relates to other fold switching proteins. This focus is appropriate. However, much of the Discussion has an introductory nature (could some parts be moved in Introduction?). Connections to previous work is limited to two sentences at the end of each of the two paragraphs and are rather general.

Response: We thank the reviewer for pointing this out. We have revised the Discussion section as suggested, focusing on the relationships between the fold switch and the enzyme mechanism.

Minor comments:

1) Figures 1 and 2 need some work. It should be noted in Figure 1 legend that the regulatory domain 377-447 is absent from the apo state. In a), secondary structure elements for both states should be included. This would greatly help to illustrate the structural changes that take place. Much of Figure 2 appears unnecessarily shaded. For example, features highlighted by boxes should not be shaded at all (it appears counterproductive to me). Colors in Figure 2 appear to be the same as in Figure 1, but mean different things. This should be avoided.

Response: We have redrawn Figure 1 and 2 as suggested.

2) The term “fold-switching motif” is used for the first time on line 160 and thereafter repeated throughout the text. For clarity, at first mention, specify which residues make up this motif.

Response: We thank the reviewer for pointing this out. The fold-switching motif consists of residues 351-375. We have added this information in the revised manuscript.

3) Why does the peak at 1657 cm^{-1} in the Raman spectrum disappear completely after incubation with InsP6 when much of the alpha helical structure remains in the structure?

Response: The Raman spectrum in this study was determined using surface-enhanced Raman scattering (SERS) technique, in which effective Raman signals can be detected when Au@AgNPs closely interact with proteins. So the peaks observed in the experiment are the

reflection of those secondary structures close to the bound Au@AgNPs, but not the contribution of the whole protein. We find several amide groups (Asn 296, Asn298, Asn 348 and Gln360) and one thiol group (Cys307) near or on α F that are exposed to the solvent and maybe helpful to bind the nanoparticles. It is highly possible that the peak at 1657 cm^{-1} is contributed by the α F-Au@AgNPs interactions. While in the structure of PopP2-InsP6 complex α F is replaced by a β -hairpin, so this Raman peak disappeared.

4) On line 176, a region between the two basins of attraction has been designated “transition state”. I think this is too strong. No evidence is presented that the selected region satisfies suitable criteria for a transition state (e.g., that this point kinetically partitions the two basins).

Response: We thank the reviewer for pointing this out. We revised it as “during the transition process”.

5) On line 184 “To test this possibility, we firstly performed a 3x500 ns all-atom molecular dynamics (MD) simulations for apo and InsP6-bound PopP2”. Presumably, it is meant 3x500 ns for *each of* the two structures? Please specify.

Response: We have changed the sentence to “To test this possibility, we performed 3×500 ns all-atom molecular dynamics (MD) simulations for apo and InsP6-bound PopP2, respectively.”

6) The RMSD values reported on line 189 should be clarified. Are they the values at the end of the simulations? Averages taken over part of simulations?

Response: It is an average RMSD for the final 100 ns MD simulations. We clarified this point in the main text accordingly (Line 215).

7) Line 193 “calculation of binding energy”. Specify the method used.

Response: Molecular-mechanics/generalized Born surface area (MM/GBSA) method was used to calculate binding energies of RRS1_{WRKY} to the InsP6-bound PopP2 and the apo PopP2. We have added this information in the revised manuscript (Line 219-221).

8) The conclusion that the fold switches goes via a relatively disordered intermediate conformational ensemble is interesting. The behavior is similar to that found for the C-terminal domain of the RfaH protein (see, for example, Biopolymers. 2021;e23420 or J. Phys. Chem. B 2014, 118, 5101). This can be worth pointing out.

Response: We thank the reviewer for the advice and have revised the Discussion as “In this study, we found that a fold switch happens in a small motif located between the regulatory domain and the substrate binding helix of PopP2, serving as a transition system to relay the InsP6 binding signal from the regulatory domain to the substrate binding helix. A random coil is revealed during the fold switching process from a long α -helix to a β -hairpin. A similar process was also reported in the C-terminal domain of transcription factor RfaH, in which an α -helical hairpin is refolded into a 5-stranded β -barrel through an unfolded state. It is worth investigating the universality of this process in similar fold-switching proteins” (Line 381-389).

9) On line 420, specify the Schrödinger software (they have many products).

Response: We used the Protein Preparation Wizard in Schrödinger software. We have added this information in the revised Method (Line 572).

10) Regarding MM/GBSA calculations, it would be nice if the authors could present a dissection of the binding energy in terms of enthalpy and entropy components, as is often done for this kind of binding affinity calculations. Specifically, the PCA analysis suggests a much more flexible RRS1_{WRKY} binding region in the apo state than the InsP6 bound state. Can this be confirmed by different Delta S components of the binding energy between the two states?

Response: The enthalpy and entropy components of the binding energies are shown in Table S2.

Table S2. Energies of RRS1WRKY binding to apo- and InsP6-bound PopP2. All terms are in kcal/mol

	ΔG (kcal/mol)	ΔH (kcal/mol)	$-T \Delta S$ (kcal/mol)
apo-PopP2	-3.8 ± 3.8	-59.7 ± 5.9	55.9 ± 8.7
InsP6-bound PopP2	-9.1 ± 4.3	-85.3 ± 8.1	76.2 ± 10.5

The ΔS term for the binding energy is calculated according to

$$\Delta S_{\text{apo}} = S_{\text{Complex,apo}} - S_{\text{Receptor,apo}} - S_{\text{Ligand,apo}}$$

$$\Delta S_{\text{holo}} = S_{\text{Complex,holo}} - S_{\text{Receptor,holo}} - S_{\text{Ligand,holo}}$$

If we want to compare the flexibility of the binding region in the apo state and that in the InsP6 bound state, the ΔS component we need to obtain is $S_{Receptor (binding region),holo} - S_{Receptor (binding region),apo}$, which cannot be obtained with the MM/GBSA calculation of RRS1_{WRKY} to the apo and InsP6 bound PopP2.

Reviewer #2 (Remarks to the Author):

The authors present a paper describing the mechanism of allosteric activation by inositol hexaphosphate (InsP6) of a YopJ T3SS effector family member (PopP2) from the plant pathogen *Ralstonia solanacearum*. They present for the first time an Apo structure of PopP2 (not bound to InsP6). Comparison of the Apo structure to that of other PopP2-InsP6 Complexes revealed binding in a remote site to dictate a conformation switch in the catalytic site that also stabilized the substrate recognition helix. The authors support the structure observations with biochemical and computational data and provide a comparison to other YopJ family members with solved structures. Although the structure (with caveats of its instability and poor structure validation metrics) depicts an interesting fold switch, the paper describing previous PopP2 structures (PMID: 28737762) already reported similar disordered segments in MD simulations in the absence of InsP6.

Response: We thank the reviewer for the constructive suggestions, which will significantly improve the manuscript. In the paper mentioned by the reviewer (PMID: 28737762), we presented a MD simulations study on the conformational changes of PopP2 bound to InsP6, AcCoA and RRS1-R_{WRKY}, sequentially. The result indicated a moderate change of r. m. s. f. of the atomic fluctuation in the fold-switching motif discovered in this paper. But we didn't pay attention to this motif at that time, due to lack of evidence to link it with substrate binding of PopP2. In this study, we identified that this motif works as a transition system to relay the InsP6 binding signal from the regulatory domain to the substrate binding helix by shuffling its secondary structures. This discovery is unexpected and never reported in any previous publications about the YopJ family effectors.

We understand the reviewer's concern about the structure validation metrics and further refined the structure. The Rfree has been decreased from 0.245 to 0.232, the Clashscore from 11 to 3 and the RSRZ outliers from 7.3% to 5.3%. We tried very hard to improve the model, but could not get a lower Rfree. It might be because that about 1/3 of the residues are either missing or highly flexible in the apo PopP2 structure.

More importantly, the authors do not discuss how the switch relates to the enzyme mechanism (and some of their experiments were designed without its' consideration). Instead, the authors provide a somewhat lengthy discussion of fold change, listing several random examples where switching occurs without appropriate reference to the rich history of structural plasticity that had been described for decades.

Response: We thank the reviewer for pointing this out and agree with the reviewer that the Discussion part was too general and our understanding about the structural plasticity is limited. We have revised this part as suggested, focusing on the relationship between the fold switch and the enzyme mechanism.

The description of the active site in the introduction is confusing, which residues contribute to the acetyltransferase activity of PopP2?

Response: We thank the reviewer for pointing this out. The residues in the active site of PopP2 have been stated in the introduction (Line 129-130) and labeled in Figure 1b.

Figure 2 is difficult to see the conformation changes. Perhaps using different colors for the parts of the structure that move? For example, the light blue apo structure could have blue for the parts described in the text as having a conformation change and the InsP6 could have red.

Response: We have redrawn Figure 2 using different colors to show the structures that move. We also labeled those structures in Fig 2a to make it easier to see where the structural changes take place.

In Figure3B, the minima look the same to me. Is the difference between the apo and InsP6 bound state significant? Why is it "as expected" that the global minima would be for the apo state?

Response: There are around 3 kcal/mol difference between these two states. Without the presence of InsP6, the apo state should be the global minima, as it is crystalized by the experiment.

MD simulations in PMID 28737762 suggest the same region from 350-370 has increased rmsf. Can the global energy minima for the helix in the apo state be explained in light of the published rmsf for the same region in the apo state? I can not tell from the PCA arrows in figure S3 if the MD simulations recapitulate the published data.

Response: Actually, the region from 350-375 has a decreased rmsf after PopP2 binds with InsP6 in PMID 28737762, as you can see in the attached figure below (PopP2 in apo state is colored in black, InsP6-bound PopP2 is in green).

However, it might be inappropriate to compare the results of these MD work. The rmsf of the apo PopP2 above was generated based on the InsP6-bound PopP2 structure, in which the fold-switching motif shows as a hairpin. Based on the crystal structure presented in this paper, the reported rmsf of PopP2 in PMID 28737762 should not be right.

Figure 4A looked at the binding of only two InsP6 residues (R380 and K383). What about K453, which trades a hydrogen bond with a neighboring Ser in the Apo state with a hydrogen bond to InsP6 when it binds? This residue seems relevant to the mechanism being described by the authors (instead of reporting K383 is the key residue) and should not be excluded.

Response: We are sorry that aim of these experiments was not well explained. We actually want to check the hypothesis that InsP6 regulates the substrate binding through the fold-switching motif, by introducing mutations in the InsP6 binding pocket and the fold-switching motif. So we selected two residues (R380 and K383) in the InsP6 binding pocket, both of which are located on the α -helix immediately after the fold-switching motif. We found K383A can effectively reduce the association of PopP2 and RRS1-R WRKY domain. The main reason we did not select K453 is because it is not a strictly conserved residue in the YopJ family effectors. In HopZ1a, the corresponding residue F355 is not involved in direct InsP6 recognition (*Nat. Struct. Mol. Biol.* 2016, 23(9): 847-852).

The BLI binding curves are performed in a background active site C321A mutation, which is not stated in the text. Why? Especially considering the reported ping pong mechanism of acetyltransferase activity where AcCoA binds first and acetylates this Cys prior to RRS1 binding.

Response: As demonstrated by previous studies (*PLoS Pathog.* 6(11): e1001202), PopP2 can autoacetylate on K383 in *E. coli*, thus disrupting the interaction between InsP6 and PopP2. To prevent the effect of autoacetylation, the mutants used for BLI assays were prepared on the active site C321A mutation. The same strategy has been used in our previous study on PopP2 (*Nat. Plants*, 2017, 3:17115). We have stated the reason in the manuscript (Lines 255-259).

The L369P/V370P and L371P/D372P seem quite disruptive to both the helix and hairpin conformations of the switch region. Are these mutations stable? Can they bind InsP6 or AcCoA?

Response: L369-D372 are located on the very C-terminus of α F followed by the flexible regulatory domain in the apo-PopP2 structure; in the PopP2-InsP6 structure, there is still a long loop between D372 and the first InsP6 binding residue in the regulatory domain. So these mutations should not affect the InsP6 and AcCoA binding. Furthermore, all the mutants used in this study can be overexpressed in *E. coli* with high yield, and behave similar to the WT PopP2 on chromatography. We also observed the *in planta* accumulation of the different PopP2 variants either transiently expressed via *Agrobacterium* or delivered by *Pseudomonas fluorescens* (Fig. S5 and S9). These results suggest these mutations do not affect the overall conformation of the proteins.

The crude acetylation assay in figure 4c does not measure acetylation, it measures binding to DNA. The statement “unacetylated RRS1 proteins do not form complex with W box” is not shown anywhere in the paper.

Response: We thank the reviewer for pointing this out. Following the reviewer’s suggestion, we both tested the acetylation status of RRS1-R WRKY domain *in vitro* as well as *in planta* (Fig 4a and 4c). Our data now clearly show that RRS1 WRKY domain can be acetylated by WT PopP2 and K380A mutant, which is consistent with the result of DNA binding assay in which only unacetylated RRS1_{WRKY} interacts with DNA (Fig 4b). We also clarify in the revised manuscript that acetylation of RRS1_{WRKY} leads to disruption of RRS1_{WRKY}-DNA interaction (Line 86-88).

Figure 5 has typos in the figure legend (wilt for wild) and experiments in 5a should be performed in triplicate for acceptable rigor.

Response: We thank the reviewer for pointing this out. We have revised the typos and, as requested, the experiments shown in Figure 5 have now been performed in triplicate with similar results.

Figure S3c and d have several lines of colors that are not defined in the figure legend.

Response: We thank the reviewer's comments. Each simulation was repeated three times which were represented by green, blue and orange lines. We have added the information about the lines in the figure legend of Figure S3c and S3d.

Although the name might be relatively new and coined in the referenced review, the concept of “fold switching” is not recent. The term “switch” was described in 1990 for changes in active/inactive ras protein structures. Similar studies describe chameleon sequences (2015) in homologs that reflect structural plasticity. Also reviews since 2002 describe the plasticity of protein folds PMID: 12127461.

Response: We agree with the reviewer that the concept of “fold switching” is not something quite new. Fold switching is a process including remodeling of secondary structure in response to a few mutations (evolved fold switchers) or cellular stimuli (extant fold switchers). With the quick expansion of the Protein Data Bank (PDB), more and more proteins are reported to change their conformations. However, an exhaustive search of PDB only identified about 96 extant fold switchers (*PNAS*. 2018, 115(23):5968-5973). A computational method has been therefore developed to explore the PDB and yielded more potential fold switching proteins, but PopP2 is not in the list, suggesting that it is still difficult to predict the fold switching proteins based on the knowledge we know about it. We believe this work provide a new tool to study fold switching proteins, as well as structural plasticity.

Line123 well reserved?

Line 127 ahead of

Line 145 change lack of to lacking

Line 154 the regulatory

Line 169 simulation, which

Line175 with what was observed

Line 169-170 path collective variables

Line 190 switch spelling

Line 286 to from?

Line 289 represent

Response: We have corrected these errors in the revised manuscript.

Reviewer #3 (Remarks to the Author):

This article by Yao Xia et al. addresses the molecular mechanism of inositol hexaphosphate (Insp6)-mediated allosteric regulation of YopJ family acetyltransferases. The manuscript highlights the previously undescribed fold-switching mechanism that regulates the substrate binding of YopJ family acetyltransferases by comparing the crystal structure of PopP2, a YopJ family type III secreted effector in apo- and Insp6-bound state. Furthermore, using biochemical and computational analysis the authors further demonstrated the chain reaction of conformational changes induced by the interaction of Insp6 with PopP2. The manuscript is well written and provides interesting findings underlying allosteric regulation of YopJ family acetyltransferases. However, there are few weaknesses.

Response: We are grateful to the reviewer for his/her positive comments on our manuscript. Our responses to the reviewer's advice are listed below.

Specific comments: The authors demonstrated how Insp6 regulates substrate binding of PopP2 through fold switching motif. To address this, the authors used various biochemical and biophysical approaches to investigate the role of residues in the Insp6-binding pocket and fold-switching motif in mediating the interaction of PopP2 with RRS1WRKY. Using bio-layer interferometry (BLI) assay and in vitro acetylation assay, the author highlighted the significance of K383A (in Insp6 binding pocket) and L369-D372 (in fold-switching motif), in regulating the interaction between PopP2 and RRS1WRKY. However, any evidence of the effect of these mutations on the overall conformation of PopP2 is missing.

Response: L369-D372 are located on the very C-terminus of α F followed by the flexible regulatory domain in the apo-PopP2 structure; in the PopP2-Insp6 structure, there is still a long loop between D372 and the first Insp6 binding residue in the regulatory domain. Therefore, these mutations should not affect the Insp6 and AcCoA binding. Furthermore, all the mutants used in this study can be overexpressed in *E. coli* with high yield, and behave similar to the WT PopP2 on chromatography. We also observed the accumulation of the different PopP2 variants *in planta*, either transiently expressed via *Agrobacterium* in *N.*

benthamiana (Fig 4c) or delivered by *Pseudomonas fluorescens* in *Arabidopsis* (Fig. S9). These data suggest these mutations do not affect the overall conformation of the PopP2 proteins.

Figure 4(c): It would be good to add additional panel (either in main figure or as a supplementary image) displaying the acetylated state of RRS1WRKY by PopP2 wild type and variants in the presence of InsP6, AcCoA.

Response: We thank the reviewer for pointing this out. Following the reviewer's suggestion, we performed western blot to test the acetylation status of RRS1-R WRKY domain *in vitro* as well as *in planta* (Fig 4a and 4c). Together, our data clearly show that RRS1-R WRKY domain is acetylated only by WT PopP2 and K380A mutant, consistent with the result of DNA binding assay.

Figure 5(b): The PopP2 mutants mentioned in the figure legend for the ion-leakage assays seem incorrect. It should be L369P/V370P and 371P/D372P instead of L369A/V370A and 371A/D372A.

Response: We have corrected the errors in the revised the manuscript.

Minor comments:Line 588 - wilt > wild

Line 190 - swithc > switch

Line 169-170 - path collective various > path collective variables

Figure S2: Figure legend – color of InsP6-bound PopP2 should be pink not red.

Response: We have corrected these errors in the revised the manuscript.

Reviewer #4 (Remarks to the Author):

The findings of the study clearly demonstrates a long-suspected role of InsP6 in making PopP2 (and other acetyltransferase effectors) competent for substrate binding, particularly through a novel fold-switching modification. The study is of some interest to the broader scientific community and findings may be applicable for future development of drugs that could target the co-factor binding ability of this effector family from plant and animal pathogens. I feel that the scope of this manuscript does indeed suit this journal, but several issues with this manuscript will need addressing first.

Response: We thank the reviewer for his/her positive comments on this work and the constructive suggestions. We have addressed the reviewer's comments point-by-point below.

The following major issues should be addressed prior to acceptance for publication:

1. Inclusion of a phylogenetic tree of effectors from this family (including PopP2, AvrA, YopJ and HopZ1a) will be important for understanding the conclusions reached in this manuscript. This is particularly interesting if this was presented specifically for the 'regulatory domain – aa377-447' or the 'acetyltransferase domain – aa149-488' described in Figure 1. I also recommend adding corresponding regions from several non-acetyltransferase effectors from the CPD family to delineate where the broader conclusions about the YopJ family stand with regard to other CPD effectors.

Response: We thank the reviewer for this great idea. We have included in the revised manuscript a phylogenetic tree of representative proteins from CE clan of proteases, including YopJ family effectors produced by both plant and animal pathogens and three deubiquitinases (Fig S10). As suggested, the phylogenetic tree is presented specifically for the acetyltransferase domain for YopJ family effectors and the catalytic domain of the deubiquitinases. On the phylogenetic tree, the YopJ family effectors are clearly separated from other CE clan of proteases. We also performed sequence alignment, which shows that the regulatory domain only exists in the YopJ family effectors (Fig S11). These data suggest that the way of substrate regulation characterized in this study has been evolved independently in the YopJ family effectors.

2. L211-212: Authors suggest that the “L369P/V370P and L371P/D372P mutations completely abolished acetyltransferase activity (Fig 4c)” but the figure panel only demonstrates DNA binding by the RRS1 WRKY DNA-binding domain and not acetylation status of this domain. As such, this would be ideal using an anti-Acetyl Lysine antibody (α -AcK) for a western blot. This assay has not been used in this manuscript and should be included to demonstrate that acetylation is in fact affected leading to the predicted antagonism of phosphorylation required for immunity (Guo et al. 2020 CHM, DOI: 10.1016/j.chom.2020.03.008).

Response: We thank the reviewer for pointing this out. Following the reviewer's suggestion, we tested the acetylation status of RRS1-R WRKY domain *in vitro* as well as in planta (Fig 4a and 4c). Our data clearly show that this acetylation can be detected only with WT PopP2 and K380A mutant, consistent to the result of DNA binding assay in which only

unacetylated RRS1_{WRKY} interact with DNA. In addition, *in planta* acetylation of RRS1-R WRKY domain by WT PopP2 and R380A (Figure 4c) nicely correlates with their ability to trigger activation of RPS4/RRS1-R-dependent immunity (Fig 5). As mentioned by the reviewer, phosphorylation of Thr1214 and acetylation of Lys1221 in RRS1-R WRKY domain were indeed previously shown to play a mutually antagonistic role in RRS1-R activation.

3. L179-214: Authors demonstrate using molecular dynamics and BLI assays to demonstrate that RRS1-WRKY binding by PopP2 is affected by the InsP6-binding triggered fold-switch in PopP2. However, they have not shown this via a co-immunoprecipitation (co-IP) assay. Is binding in a co-IP assay affected? Is there a reduction in acetylation (assessed by α -AcK blot)? This data would be both interesting and highly relevant to this manuscript.

Response: We thank the reviewer for pointing this out. As suggested, we investigated *in planta* the physical interaction between the different PopP2 variants with RRS1-R WRKY domain. For this, we performed a FRET-FLIM assay. This approach has been successfully used previously to demonstrate the targeting of RRS1-R or its C-terminal portion by PopP2 in the plant cell nucleus (*PLOS Pathogens*, 2010, 6(11): e1001202 ; *Cell*, 2015, 161(5): 1074-1088). Consistent with our molecular dynamics and BLI assays as well as our acetylation assays performed in *E. coli* and *in planta*, we confirmed that both L369P/V370P and L371P/D372P as well as K383A mutants were unable to physically interact with the WRKY domain of RRS1-R in living plant cells (Table 1 and Fig S7).

4. L260-262: Authors report that the B-factors for loops associated with the substrate-binding domain are higher than for the catalytic core. These values (comparison between both) are not reported and not demonstrated in a scale bar in Supplementary Fig. S6. Furthermore, is the conclusion about the YopJ family effectors (L261-264) correct regarding flexibility of these interfaces in the absence of InsP6 when they have not been measured under these conditions, but rather have apparently only been calculated for InsP6-bound structures?

Response: We thank the reviewer for pointing this out. We realize that B-factors for the loops associated with the substrate-binding domain in the InsP6-bound structures are indeed not direct evidences supporting the flexibility of them in the absence of InsP6. Therefore we removed the B-factor putty presentation from the revised manuscript.

The manuscript is generally well written, apart from a number of grammatical and spelling errors outlined below:

L27: “employed in” is strange terminology. Perhaps “deployed through” is better.

L39: “form target” > “form a target”

L51: “to rapid” > “to the rapid”

L54: “family which are” > “family, whose members are”

L62: “protease” > “proteases”

L63: “amounting” > “mounting”

L67: “targets” > “target”

L70: “to suppressed” > “to a suppressed”

L77: “root infecting” > “root-infecting”

L82-83: This sentence is very vague. What “about the modification sites” is unclear?

L84: “between YopJ” > “between the YopJ”

L120: “unable to build, due” > “unable to be built due”

L123: “well reserved” – do you mean “well-conserved”?

L145: “constrained, lack” > “constrained, with lack”

L149: “in recognition” > “in the recognition” – should ‘recognition’ instead be ‘interaction’?

L154: “regulatory domain” > “the regulatory domain”

L158: “and regulatory” > “and the regulatory”

L164: “in amide” > “in the amide”

L165: “ α -helix” > “the α -helix”

L170: “various” > “variables”

L171: “ α -helix into β -strand” > “an α -helix into a β -strand”

L175: “what observed” > “what is observed”

L176: “in to” > “into”

L179: “Insp6” > “InsP6”; “substrate binding” > “substrate-binding” (the latter change needs to be made throughout the manuscript, not listed here further)

L180: “InsP6 binding” > “InsP6-binding”

L184: “performed a 3x500” > “performed 3x500”

L187: “that the apo PopP2” > “that apo PopP2”

L190: “switch” > “switch”

L191-192: “leading to relatively smaller and deeper pocket which” > “leading to a relatively smaller and deeper pocket, which”; “facilitate a stronger” > “facilitate stronger”

L195: “-85.3 kcal/mol vs -59.7 kcal/mol” > “-85.3 kcal/mol vs -59.7 kcal/mol, respectively”

L196: “assay” > “assays”

L197: “assay” > “assays”

L204: “between InsP6 binding pocket and substrate” > “between the InsP6 binding pocket and the substrate”

L210: “K383A mutation” > “the K383A mutation”

L215: “The PopP2-triggered” > “PopP2-triggered”

L217: “RRS1-R WRKY” > “the RRS1-R WRKY”

L228: “cell death response” > “the cell death response”

L235: “single K383A mutation” > “the single K383A mutation”

L247: Remove “members”

L286: Unclear grammatical error, please review

L294: Unclear grammatical error, please review

L297-298: “believed to widespread” > “believed to be widespread”

Response: We have corrected these errors in the revised manuscript.

REVIEWER COMMENTS

Reviewer #1 (Remarks to the Author):

The authors have carefully addressed each of the concerns raised in my report. Specifically, the simulation methods and techniques utilized are now described in much more detail. The addition of an example trajectory as a movie is also helpful. Also, the discussion section has also been improved. A minor stylistic error might need to be fixed on line 353, changing "Such as" to "For example". With the (substantial) changes made by the authors, my recommendation is that the manuscript is accepted for publication.

Reviewer #3 (Remarks to the Author):

We've looked over the revised submission and are happy that the authors have satisfactorily addressed our critique of the first submission

Reviewer #4 (Remarks to the Author):

This revised manuscript has addressed all of my concerns and is suitable for publication. I would like to thank the authors for their excellent work in doing all the requested experiments and the high quality of their research.

We thank all the reviewers for their positive comments on our work. Please find our point-by-point response to each of the reviewers' comments below.

REVIEWER COMMENTS

Reviewer #1 (Remarks to the Author):

The authors have carefully addressed each of the concerns raised in my report. Specifically, the simulation methods and techniques utilized are now described in much more detail. The addition of an example trajectory as a movie is also helpful. Also, the discussion section has also been improved. A minor stylistic error might need to be fixed on line 353, changing "Such as" to "For example". With the (substantial) changes made by the authors, my recommendation is that the manuscript is accepted for publication.

Response: We thank the reviewer for recommending the publication of our paper. The manuscript has been revised as suggested.

Reviewer #3 (Remarks to the Author):

We've looked over the revised submission and are happy that the authors have satisfactorily addressed our critique of the first submission

Response: We thank the reviewer for the positive comments on our manuscript.

Reviewer #4 (Remarks to the Author):

This revised manuscript has addressed all of my concerns and is suitable for publication. I would like to thank the authors for their excellent work in doing all the requested experiments and the high quality of their research.

Response: We thank the reviewer for the positive comments on our manuscript.